# N-terminal α-amino SUMOylation of cofilin-1 is critical for its regulation of actin depolymerization

Weiji Weng[1,6], Xiaokun Gu[1,6], Yang Yang[1,6], Qiao Zhang[1], Qi Deng[1], Jie Zhou[1], Jinke Cheng [1], Michael X. Zhu [2], Junfeng Feng [3,4] ✉, Ou Huang [5] ✉ & Yong Li [1] ✉

Small ubiquitin-like modifier (SUMO) typically conjugates to target proteins through isopeptide linkage to the ε-amino group of lysine residues. This posttranslational modification (PTM) plays pivotal roles in modulating protein function. Cofilins are key regulators of actin cytoskeleton dynamics and are well-known to undergo several different PTMs. Here, we show that cofilin-1 is conjugated by SUMO1 both in vitro and in vivo. Using mass spectrometry and biochemical and genetic approaches, we identify the N-terminal α-amino group as the SUMO-conjugation site of cofilin-1. Common to conventional SUMOylation is that the N-α-SUMOylation of cofilin-1 is also mediated by SUMO activating (E1), conjugating (E2), and ligating (E3) enzymes and reversed by the SUMO deconjugating enzyme, SENP1. Specific to the N-α-SUMOylation is the physical association of the E1 enzyme to the substrate, cofilin-1. Using F-actin co-sedimentation and actin depolymerization assays in vitro and fluorescence staining of actin filaments in cells, we show that the N-α-SUMOylation promotes cofilin-1 binding to F-actin and cofilin-induced actin depolymerization. This covalent conjugation by SUMO at the N-α amino group of cofilin-1, rather than at an internal lysine(s), serves as an essential PTM to tune cofilin-1 function during regulation of actin dynamics.

Cofilins belong to the actin-depolymerizing protein family, which consists of cofilin-1 (CFL1, n-cofilin, non-muscle), cofilin-2 (CFL2, muscle cofilin), and actin-depolymerization factor (ADF, destrin) and is well conserved among eukaryotes[1]. These proteins have a molecular mass of 15–19 kDa and share multiple structural motifs, including an actin-depolymerizing factor homology (ADF-H) domain that allows for binding to actin, a central alpha-helix, an N-terminal extension, and a C-terminal helix[2,3]. Cofilins exert their cellular function through regulation of actin cytoskeleton dynamics[4,5]. Despite similarities in the structure, they differ in their affinity for actin and hence the efficiency in actin depolymerization[6–8]. As ADF and CFL1 are more efficient in actin depolymerization than CFL2[7], they are also mainly expressed in tissues with higher actin turnover rates. Moreover, while ADF is better at sequestering monomeric actin, CFL1 is more efficient at nucleation of and severing actin filaments[9].

[1]Department of Biochemistry and Molecular Cell Biology, Shanghai Key Laboratory for Tumor Microenvironment and Inflammation, Shanghai Jiao Tong University School of Medicine, Shanghai 200025, China. [2]Department of Integrative Biology and Pharmacology, McGovern Medical School, The University of Texas Health Science Center at Houston, Houston, TX 77030, USA. [3]Brain Injury Centre, Renji Hospital, Shanghai Jiao Tong University School of Medicine, Shanghai 200127, China. [4]Shanghai Institute of Head Trauma, Shanghai 200127, China. [5]Department of General Surgery, Comprehensive Breast Health Center, Ruijin Hospital, Shanghai Jiao Tong University School of Medicine, Shanghai 200025, China. [6]These authors contributed equally: Weiji Weng, Xiaokun Gu, Yang Yang. ✉e-mail: fengjfmail@163.com; ou_huang@126.com; liyong68@shsmu.edu.cn

Accumulating evidence has revealed that cofilin can undergo a multitude of posttranslational modifications (PTMs), such as phosphorylation, ubiquitination, neddylation, S-glutathionylation, O-GlcNAcylation, and oxidation[6,10–14]. Phosphorylation of CFL1 at Ser3 occurs dynamically and rapidly, and it serves as a key convergence point for controlling cofilin activity to regulate various signaling pathways in health and disease conditions[6,15]. As the phosphorylation at CFL1 Ser3 inhibits its binding to either F- or G-actin, the dephosphorylation makes CFL1 active[5,16]. In addition to phosphorylation, cofilins are also regulated by reversible ubiquitination at any one of their several ubiquitinable lysines[10], and by neddylation[11]. Moreover, S-glutathionylation of cofilin reduces its ability to depolymerize F-actin, which recovers after dethionylation[12]. Oxidation of the thiol groups of the conserved cysteine residues (Cys39 and Cys80) in cofilins leads to the formation of an intramolecular disulfide bond, which although weakens ADF/cofilin phosphorylation, still suppresses their F-actin depolymerizing activity[17]. Oxidation of CFL1 at the non-conserved Cys139 and Cys147 was also reported to inhibit the ability of cofilin to sever actin filaments, which contributes to adhesion and helps maintain directional migration[14]. O-GlcNAcylation represents a unique PTM of CFL2, which may allow its function to be independently regulated from that of ADF and CFL1[13], likely for its involvement in diseases where deregulation of O-GlcNAcylation takes place[18].

PTM by the small ubiquitin-like modifier (SUMO) also plays important roles in various cellular processes by altering the stability, conformation, interactions, and/or subcellular localization of target proteins. Here, we report that CFL1 is SUMOylated at its N-terminal α-amino group and this modification plays a critical role in actin depolymerization. Previously, SUMO conjugation has only been known to occur on ε-amino groups of internal lysine residues of the target proteins. Our finding provides the first evidence that SUMOylation at the N-terminal α-amino group of a protein, rather than the internal lysine(s), is essential and sufficient for controlling the protein's key function.

## Results

### CFL1 is modified by SUMO1 in vivo and in vitro

To test whether CFL1 is a SUMO substrate, we transiently transfected Chinese hamster ovary-K1 (CHO-K1) cells with the plasmid expressing C-terminal hemagglutinin (HA)-tagged mouse CFL1 (CFL1-HA), together with or without the plasmid for N-terminal His-tagged SUMO1 (His-SUMO1). Lysates were subjected to denaturing immunoprecipitation (De-IP) with anti-HA antibody followed by Western blotting with anti-SUMO1 antibody. As shown in Fig. 1a, a band with a molecular mass of ~37 kD appeared when CFL1 was coexpressed with SUMO1. The ~19 kD increase in mass of the predicted molecular weight of CFL1 (~18

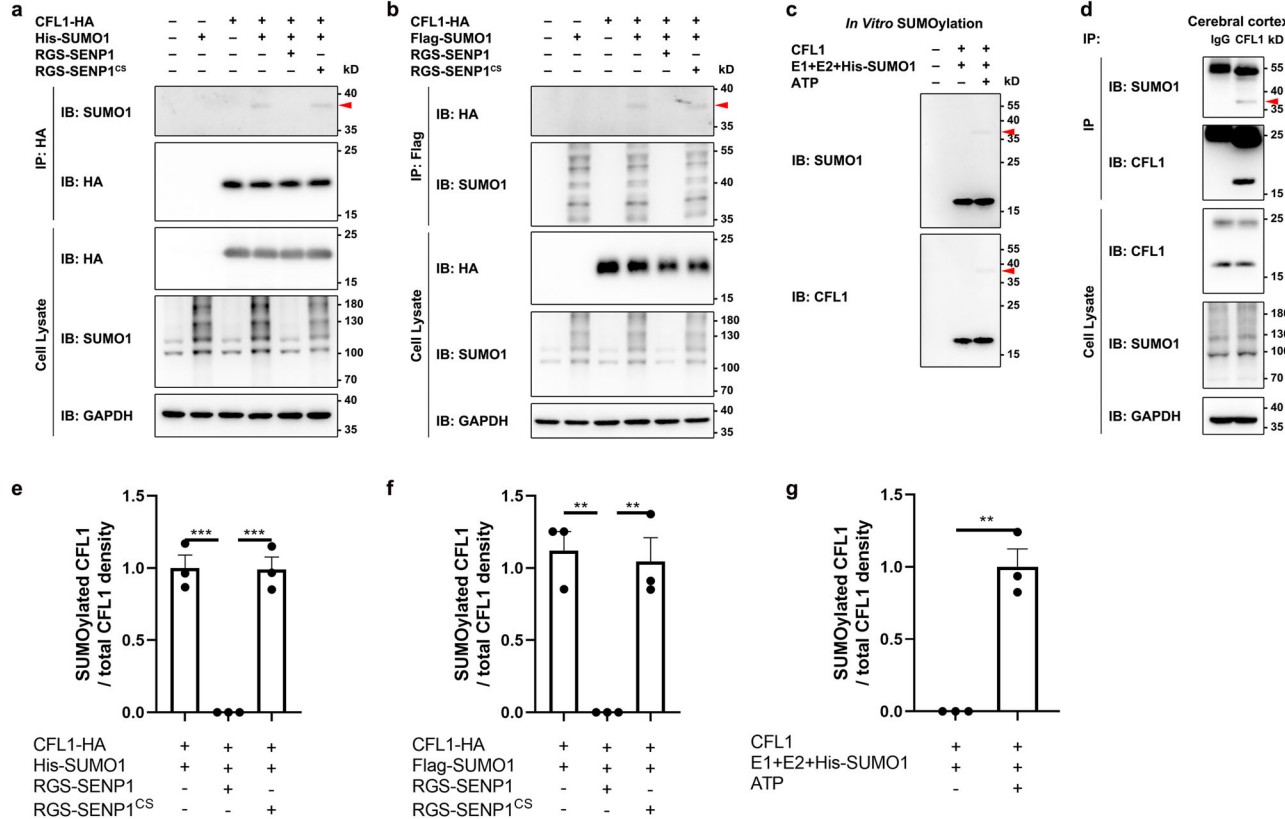

**Fig. 1 | CFL1 is modified by SUMO1. a, b** CFL1 is SUMO1-conjugated when expressed in CHO-K1 cells. Lysates from CHO-K1 cells transiently transfected with empty vector (−), CFL1-HA, His-SUMO1 (**a**) or Flag-SUMO1 (**b**), RGS-SENP1, and RGS-SENP1CS at various combinations as indicated for 24 h were subjected to De-IP with the anti-HA (**a**) or anti-Flag (**b**) antibody, which was followed by immunoblotting (IB) using anti-SUMO1 and anti-HA antibodies. The original lysates were analyzed by IB with anti-HA and anti-SUMO1 antibodies for input, and anti-GAPDH antibody for loading control. Red arrows indicate SUMOylated CFL1. **c** CFL1 is SUMO1-conjugated in vitro. Purified recombinant CFL1 was incubated with E1, E2, SUMO1, and adenosine triphosphate (ATP) at various combinations as indicated in vitro at 37 °C for 1 h, and the reaction was terminated with SDS loading buffer. The samples were analyzed by IB with anti-SUMO1 and anti-CFL1 antibodies. Red arrows indicate SUMOylated CFL1. **d** SUMO1 conjugation of endogenous CFL1 in mouse cerebral cortex in vivo. Lysates prepared from mouse cerebral cortices under denaturing conditions were subjected to IP with anti-CFL1 antibody, followed by IB with anti-SUMO1 and anti-CFL1 antibodies. The original lysates were analyzed by IB with anti-SUMO1 and anti-CFL1 antibodies for input, and anti-GAPDH antibody for loading control. Red arrow indicates endogenous SUMOylated CFL1. **e**–**g** The quantification of Western blots of (**a**)–(**c**), respectively. Mean ± SEM; **p < 0.01, ***p < 0.001, by one-way ANOVA with Tukey's multiple comparison test for (**e**) and (**f**), and by two-tailed t test for (**g**). All blots represent ≥3 independent experiments. The exact p value and source data are provided as Source Data file.

kD) indicated that CFL1 was covalently conjugated by SUMO1. SUMOylation is known to be a highly dynamic PTM, and the abundance of SUMOylated proteins is typically quite low at any giving time[19–21]. Therefore, the intensity of the SUMOylated CFL1 band in Western blot was relatively weak. For this reason, the SUMOylated CFL1 was difficult to see in Western blot by anti-HA antibody for the immunoprecipitated samples unless when prolonged exposure was used, which showed much weaker intensity than the unconjugated CFL1 (Supplementary Fig. 1a). Additionally, a faint band indicating SUMOylated CFL1 was discernible with the use of the anti-His antibody (Supplementary Fig. 1c). In reciprocal experiments using cells coexpressing CFL1-HA and Flag-SUMO1, we also detected SUMO-conjugated CFL1 by De-IP with anti-Flag followed by Western blotting using anti-HA (Fig. 1b). More importantly, in both experimental settings, the SUMOylated CFL1 was abolished by the co-expression of arginine-glycine-serine-tagged SENP1 (RGS-SENP1), a sentrin-specific protease that deconjugates SUMO, but not the catalytically inactive mutant of SENP1 (RGS-SENP1$^{CS}$) (Fig. 1a, b, e, f). These data confirm that CFL1 is conjugated with SUMO1 and SENP1 is involved in CFL1 deSUMOylation.

Using purified recombinant CFL1 proteins, we performed in vitro SUMOylation assay and detected SUMOylated CFL1 in the presence of SUMO-conjugating enzymes and SUMO1 only when ATP was added, but not when it was omitted (Fig. 1c, g). Furthermore, using mouse cerebral cortex lysates for De-IP by anti-CFL1 antibody followed by Western blotting with anti-SUMO1, we detected a band that was not pulled down by the nonspecific IgG and had the size predicted for SUMO1-conjugated CFL1 (Fig. 1d), demonstrating the endogenous presence of SUMOylated-CFL1 in the mouse brain. Taken together, these results suggest that CFL1 is conjugated with SUMO1 in vitro and in vivo.

## The α-amino group at the N-terminus is the primary site of CFL1 SUMOylation

It is well known that SUMO conjugation occurs on internal lysine residues of target proteins. Mouse CFL1 contains 25 lysines, out of its total of 166 amino acids (Supplementary Fig. 2a). Among them only one, K132, adheres to the SUMO consensus motif, ψ–K–X–E, where ψ is hydrophobic and X is any amino acid[22]. However, mutation of K132 to arginine, K132R, did not alter CFL1 SUMOylation (Supplementary Fig. 2b). In fact, individual K → R substitutions did not reveal a single mutation able to abolish CFL1 SUMOylation, although a dramatic increase was seen with K34R and marked decreases were found with K112R and K114R mutants (Supplementary Fig. 2b). We considered the possibility that both K112 and K114 were SUMOylated by testing the double substitutions, K112R/K114R or 2KR. Although 2KR decreased CFL1 SUMOylation further than the individual substitutions (Supplementary Fig. 3a, b), it did not abolish it. Unexpectedly, mutating K112 and K114 to glutamines (2KQ) resulted in increased instead of decreased SUMOylation (Supplementary Fig. 3c), suggesting that it is unlikely that these two lysine residues are directly conjugated by SUMO. Rather, they may modulate SUMOylation at another site.

To identify the SUMOylation site of endogenous mouse CFL1, we performed LC-MS/MS analysis on endogenous CFL1 proteins purified by De-IP from mouse-derived Neuro-2a cells transiently transfected with His-SUMO1$^{E93R}$. The E93R substitution allows SUMO1 to be cleaved by trypsin to generate a QTGG-tag to the SUMO acceptor sites on the substrate (Fig. 2a). Surprisingly, we only detected a QTGG-conjugated N-terminal peptide (Fig. 2b), but not any QTGG-tagged lysines, from the endogenous CFL1. This suggested that the α-NH$_2$ group at CFL1 N-terminus, rather than the ε-NH$_2$ group of an internal lysine, is conjugated by SUMO1. This finding was unexpected since all previous studies on SUMOylation have only shown conjugation on lysine residues. On the other hand, precedent exists in case of ubiquitination, another PTM that mainly occurs in internal lysine residues, where

conjugations at the N-terminal α-NH$_2$ group, although rare, have also been reported[23].

To confirm that CFL1 is indeed conjugated by SUMO1 at its N-α-NH$_2$ group, we used both in silico and in vitro experimental approaches. First, we used AlphaFold 2 (AF2) to predict the structure of a SUMO-conjugated CFL1. AF2 estimates errors in modeled structures via both the predicted local distance difference test (pLDDT) scores and the predicted aligned error (PAE). We found that among 5 predicted models, only model 1, in which SUMO1 C-terminus is juxtaposed to the CFL1 N-terminus (Supplementary Fig. 4a), had a relatively high pLDDT score and low inter-chain PAE values (Supplementary Fig. 4b, c), supporting the likelihood of SUMO1 conjugation to the CFL1 N-terminus.

Second, we generated a lysine-less CFL1 mutant by changing all lysines to arginines (25 KR). As shown in Fig. 2c, the coexpression of 25 KR with His-SUMO1 still led to the formation of the SUMOylated band, which was further enhanced by the coexpression of the E2-conjugating enzyme, Ubc9. This finding indicates that the SUMOylation of CFL1 occurs independently of its internal lysines, showing the sufficiency of the N-α-NH$_2$ group in CFL1 SUMOylation. Conversely, we protected the N-α-NH$_2$ group by overexpressing Naa60, an N-α-acetyltransferase that specifically mediates acetylation at the N-terminus of a substrate in an irreversible manner. Coexpression of Naa60 with CFL1-HA and His-SUMO1 abolished CFL1 SUMOylation (Fig. 2d), indicating that the exposed N-α-NH$_2$ group is essential for SUMO conjugation of CFL1. Since the lysine residues of CFL1 were not protected or substituted, this result also suggests that none of the internal ε-NH$_2$ groups of CFL1 is SUMOylated.

To confirm that the N-terminus of CFL1 was indeed protected by acetylation in Naa60 overexpressing cells, we performed LC-MS/MS analysis but surprisingly found that the acetylation occurred at the second alanine instead of the first methionine of CFL1 (Supplementary Fig. 5a). As the first methionine can be cleaved, this could result from normal processing of heterologously expressed CFL1 by CHO-K1 cells; however, it was possible that the first methionine could be necessary for SUMOylation. To address this question, we took the advantage that in some cases, valine can function as an initiating amino acid by mutating M1 of CFL1-HA to a valine (CFL1$^{M1V}$-HA)[24]. We confirmed that the protein was expressed after transfection in CHO-K1 cells and when coexpressed with His-SUMO1, a high molecular weight band appeared after De-IP by anti-HA followed by Western blotting with anti-SUMO1 (Supplementary Fig. 5b). Therefore, SUMO1 conjugation occurs at the N-α-NH$_2$ group of CFL1 no matter it is a methionine or another amino acid.

Next, we validated the N-α-NH$_2$ group as the actual site of CFL1 SUMOylation in in vitro SUMOylation assays using two methods. First, we protected either all primary amine groups (Me$^{M1+25K}$-CFL1) or just the ε-NH$_2$ groups of the lysines (Me$^{25K}$-CFL1) of the purified recombinant CFL1 protein by methylation before subjecting it to in vitro SUMOylation. Mouse CFL1 protein with a 6xHis-SUMO3$^{GG}$ fused to its N-terminus was purified from E. coli. While Me$^{M1+25K}$-CFL1 was made by performing in vitro methylation after the cleavage of the SUMO3 tag by digestion with SENP2, Me$^{25K}$-CFL1 was obtained by doing methylation first and then the SENP2 digestion. The latter procedure created a free α-HN$_2$ group at the first methionine (Fig. 2e). The correct modifications were confirmed by LC-MS analysis, showing mass shifts of 731.9 and 703.5 Da for Me$^{M1+25K}$-CFL1 and Me$^{25K}$-CFL1, respectively, from the unmodified CFL1 (Supplementary Fig. 5c). Importantly, the in vitro SUMOylation assay revealed that although SUMO1 conjugated Me$^{25K}$-CFL1 similarly as the unmodified CFL1, it failed to conjugate Me$^{M1+25K}$-CFL1, further supporting that SUMOylation occurs at the N-α-NH$_2$ group of CFL1 (Fig. 2f).

In the second method, we made a recombinant N-terminal GST-tagged CFL1 protein (GST-CFL1), which contains a thrombin cleavage site between GST and CFL1. Both the untagged CFL1 and GST-CFL1 were subjected to in vitro SUMOylation first and then followed by

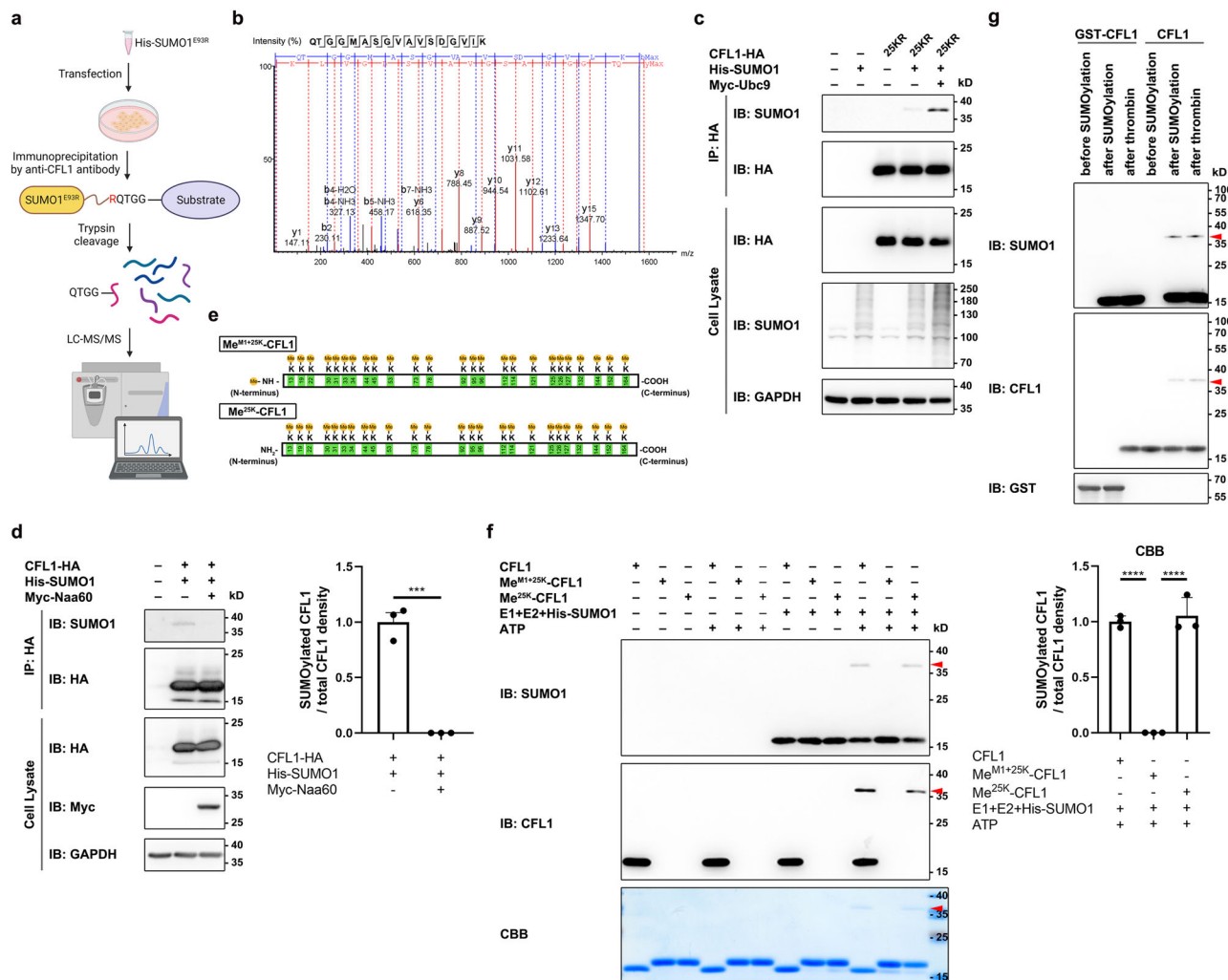

**Fig. 2 | The N-terminus of CFL1 but not internal lysine is conjugated by SUMO1.**
**a** Diagram illustrating CFL1 SUMOylation identification using IP-MS. Endogenous CFL1 proteins were purified from Neuro-2a cells expressing His-SUMO1$^{E93R}$ through De-IP with anti-CFL1 antibody. IP products underwent trypsin digestion and LC-MS/MS analysis. **b** MS/MS spectrum of a tryptic peptide containing glutamine-threonine-glycine-glycine (QTGG) preceding the N-terminal methionine of CFL1, determined by collision-activated dissociation (CAD). **c** Internal lysines not required for CFL1 SUMOylation. Lysates from CHO-K1 cells, transfected with CFL1$^{25KR}$-HA, His-SUMO1, and Myc-Ubc9, were subjected to De-IP and IB. **d** (left) N-terminal acetylation inhibits SUMOylation. Lysates from CHO-K1 cells transfected with CFL1$^{WT}$-HA, His-SUMO1, and Myc-Naa60, were subjected to De-IP and IB. (right)

Quantification of Western blots. Mean ± SEM; ***$p < 0.001$, by two-tailed $t$ test. **e** Diagram of two methylated recombinant CFL1 proteins: Me$^{M1+25K}$-CFL1 (top) and Me$^{25K}$-CFL1 (bottom). **f** (left) N-terminal α-NH$_2$ group of CFL1 modified by SUMO1 in vitro. Purified CFL1 and methylated variants were incubated with E1, E2, SUMO1, and ATP in various combinations at 37 °C for 1 h, analyzed by IB and CBB staining. (right) Quantification of CBB gel. Mean ± SEM; ****$p < 0.0001$, by one-way ANOVA with Tukey's multiple comparison test. **g** Internal CFL1 lysines were not SUMOylated in vitro. Purified CFL1 and GST-CFL1 underwent in vitro SUMOylation, followed by thrombin digestion at room temperature for 12 h (see Supplementary Fig. 5d). All blots represent ≥3 independent experiments. Exact $p$ values and source data provided in Source Data file.

digestion with thrombin to cleave the GST-tag (Supplementary Fig. 5d). While the SUMOylated band was readily detected in the untagged CFL1 protein, it was not seen in samples derived from GST-CFL1 whether or not the thrombin treatment was made (Fig. 2g). Unlike the methylated CFL1, all the lysine residues of GSF-CFL1 were not protected during the in vitro SUMOylation assay. In addition, it is unlikely that the GST tag would markedly alter the conformation of CFL1. Therefore, the lack of SUMO1 conjugation at GST-CFL1 further demonstrates that the internal residues of CFL1 are not SUMOylatable.

Taken together, the above results unveil a previously unknown form of SUMOylation at the N-terminal α-NH$_2$ group of a protein. For CFL1, this represents the only site of SUMO1 conjugation that occurs independently of the internal lysine residues, although the modification of some of the internal lysines could influence the SUMOylation efficiency at the N-α-NH$_2$ group. As described earlier, the K112R and K114R substitutions reduced whereas changing the two lysines to

glutamines, i.e., 2KQ, enhanced CFL1 SUMOylation (Supplementary Fig. 3a–c). Consistent with these residues having an allosteric effect on modulating CFL1 SUMOylation, the co-expression with Ubc9 or one of the SUMO E3 ligases such as a member of the protein inhibitor of activated STAT (PIAS) family, markedly increased the SUMOylation of the 2KR mutant (Supplementary Fig. 3b, d). In the AF2-predicted wild-type mouse CFL1 structure, K112 and K114 are located closely to the CFL1 N-terminus (Supplementary Fig. 3e). Although the 2KR mutation did not result in a marked structural change, the slightly larger size of arginine than lysine could create a spatial hindrance to the attachment of SUMO1 to the N-α-NH$_2$ group.

## N-α-SUMOylation of CFL1 uses the same machinery as conventional SUMOylation and is reversible

Protein acetylation is another form PTM that occurs both at the ε-NH$_2$ group of lysine side chains and the N-terminal α-NH$_2$ group. However,

different mechanisms are involved in their regulation, with the ε-NH$_2$ acetylation catalyzed by lysine acetyltransferases (KATs)[25], while the N-α-NH$_2$ acetylation mediated by N-α-acetyltransferases, which attach an acetyl group to the free α-NH$_2$ group in an irreversible manner since no N-terminal deacetyltransferase (NDAC) is known to exist[26]. To know if the N-α-SUMOylation of CFL1 employs the same or a different mechanism as the conventional SUMOylation, we tested its dependence on enzymes commonly involved in conventional SUMOylation. SUMOylation is a well-known ubiquitin-like conjugation process that involves an enzymatic cascade catalyzed by a heterodimeric E1-activating enzyme, an E2-conjugating enzyme, and then a ligation to the target with or without the assistance of enzyme E3; the conjugation is transient and is reversed by a family of SENPs[27,28]. Accordingly, the SUMOylation status of the substrate is regulated by a balance between SUMO E1, E2, E3, and SENPs. Thus, we knocked down E1 (SAE1/SAE2), E2 (Ubc9), and SENP1 expression individually in HEK-293T cells using small interfering RNA (siRNA) and then tested SUMOylation of endogenous CFL1 using De-IP by anti-CFL1 followed by Western blotting with anti-SUMO1. We found that the knockdown of either one of the E1 subunits (SAE1 or SAE2) or E2 diminished the SUMOylated band (Fig. 3a–c, g), whereas that of SENP1 markedly increased the SUMOylation levels of CFL1 (Fig. 3d, g). Combined with the results obtained from overexpressing Ubc9 or SENP1 (Fig. 1a, b and Supplementary Fig. 3b), these findings indicate that the N-α-SUMOylation of CFL1 employs the same set of enzymes as the conventional lysine targeting SUMOylation and is reversible.

During the conventional SUMOylation cycle, SUMO is first transferred to the catalytic cysteine of the E2 enzyme (Ubc9) from E1 (SAE1/SAE2) in the absence of the substrate. Then, the E2 (Ubc9) catalyzes formation of an isopeptide bond between the C-terminal glycine-glycine of SUMO and the designated lysine of the substrate. In this model, the E1 enzyme does not need to interact with the substrate. Indeed, under non-denaturing conditions, the anti-HA antibody did not pull down the E1 enzyme subunits, SAE1 and SAE2, from lysates of CHO-K1 cells that transiently expressed HA-tagged dual-specificity phosphatase 6 (DUSP6), a SUMO-conjugated target we reported previously[29], whether or not His-SUMO1 was coexpressed (Fig. 3e, h). By contrast, Ubc9 and SENP1 coprecipitated with DUSP6 as expected (Fig. 3e, h). Interestingly and in contrast with DUSP6, we detected, in addition to Ubc9 and SENP1, the heterodimeric subunits of the E1 enzyme SAE1 and SAE2 in the precipitants pulled down by the anti-HA antibody from lysates of cells that expressed either wild type (WT) CFL1-HA or its 25KR mutant, and the amount pulled down was increased with the co-expression of His-SUMO1 (Fig. 3e, h). We further conducted a proximity ligation assay (PLA) utilizing antibodies against endogenous CFL1, SAE1, SAE2, and SENP1 in CHO-K1 cells. The red signal, denoting protein-protein interaction, was evident when CHO-K1 cells were exposed to both SAE1 and CFL1 antibodies, thereby substantiating the interaction between CFL1 and SAE1 (Supplementary Fig. 6). Analogously, positive PLA signals were discernible for CFL1 and SAE2, as well as CFL1 and SENP1, underscoring their physical associations (Supplementary Fig. 6).

Also interesting is that while the physical association of CFL1$^{2KR}$ with E1 (SAE1/SAE2) and E2 (Ubc9) was decreased and that with SENP1 increased as compared to CFL1$^{WT}$, the association of CFL1$^{2KQ}$ with E1 and E2 was enhanced and that with SENP1 reduced (Fig. 3f, i). Thus, the 2KR and 2KQ mutations may differentially affect the binding affinities of CFL1 to the SUMO-activating, SUMO-conjugating, and deSUMOylating enzymes, resulting in opposite changes in the efficiency of SUMOylation at the CFL1 N-terminus.

## CFL1 SUMOylation enhances actin depolymerization
CFL1 is well-known for its role in actin filament non-equilibrium assembly/disassembly, involving binding to actin filaments, promoting F-actin branching, inducing filament severing, and facilitating the removal of monomeric actin[5,30]. To determine the functional significance of CFL1 SUMOylation, we first assessed how N-α-SUMOylation affects the ability of CFL1 to bind to F-actin using the actin co-sedimentation assay. This well-characterized assay uses ultracentrifugation to sediment F-actin and its binding partners. Here, we incubated purified F-actin with recombinant CFL1 proteins of wild type and various mutations before the centrifugation, and the CFL1 proteins found in the pellet and supernatant represented the F-actin-bound and unbound fractions, respectively (Fig. 4a). Consistent with the previous report[31], while CFL1$^{WT}$ bound to F-actin, the constitutively inactive mutant, CFL1$^{S3E}$, a negative control to help validate the assay, did not (Fig. 4b, Lanes 1, 2, 13, 14; and Fig. 4c).

Since the recombinant CFL1$^{WT}$ purified from *E. coli* only existed in non-phosphorylated forms, its partial association with F-actin represented the capacity of the unmodified CFL1 to bind to F-actin under this set of experimental conditions. Conversely, by mimicking phosphorylation, the S3E mutant is unable to bind to F-actin and therefore any CFL1$^{S3E}$ sedimented with F-actin represented nonspecific binding. By quantifying percent of CFL1 in pellet (%CFL1_p) against the total CFL1 in both pellet and supernatant, we found the nonspecific binding of CFL1 to F-actin to be extremely low, and without introducing SUMO, CFL1$^{WT}$ and CFL1$^{2KR}$ bound to F-actin equally well (Fig. 4c). In addition, in vitro methylation of all primary amines of the purified CFL1$^{WT}$, Me$^{M1+25K}$-CFL1$^{WT}$, did not alter the %CFL1_p either (Fig. 4b, Lanes 5, 6, and Fig. 4c), indicating that methylating all the α- and ε-NH$_2$ groups of CFL1 does not impair its binding to F-actin.

With CFL1$^{WT}$ subjected to in vitro SUMOylation, in addition to the non-modified CFL1 (indicated by blue arrow), SUMOylated CFL1 (indicated by orange arrow) was also readily detected by Coomassie Blue staining (Fig. 4b, Lanes 3, 4). Here, we quantified %CFL1_p separately for non-SUMOylated and SUMOylated bands. While for the non-SUMOylated band, the %CFL1_p remained the same as that of the untreated CFL1$^{WT}$, for the SUMOylated band, it was markedly increased by more than a half (Fig. 4b, Lanes 1, 2, 3, 4, and Fig. 4c), indicating that SUMOylation facilitates CFL1 binding to F-actin. Consistent with Me$^{M1+25K}$-CFL1 not being SUMOylatable (Fig. 2f), in vitro SUMOylation did not produce the larger sized band on the Coomassie Blue gel, ruling out any nonspecific effect from the assay per se (Fig. 4b, Lanes 7, 8); neither did it alter the %CFL1_p for the unmodified band (Fig. 4c).

To further validate that indeed, the upper band represents the SUMOylated CFL1 and it binds F-actin better than the unmodified CFL1, we engineered an N-terminal SUMO1-tagged CFL1 (SUMO1-CFL1). This fusion protein, when purified from *E. coli*, behaved just like the high molecular weight band found in CFL1$^{WT}$ subjected to in vitro SUMOylation, including the %CFL1_p value (Fig. 4b, Lanes 3, 4, 9, 10, and Fig. 4c). Given that essentially SUMO1-CFL1 possesses the same chemical structure as the N-α-SUMOylated CFL1 in terms of using an eupeptide bond to join Gly97 of SUMO1 and Met1 of CFL1 together, this result also lends a strong support to the faciliatory role of CFL1 N-α-SUMOylation in F-actin binding. Since CFL1 was 100% "N-α-SUMOylated" in this case, the observed increase in %CFL1_p (~50% above that of unmodified CFL1$^{WT}$, Fig. 4c) represented the maximal faciliatory effect of CFL1 N-α-SUMOylation on its binding to F-actin under our experimental conditions. Taken together, the above results indicate that F-actin binding of CFL1 is enhanced by its N-α-SUMOylation.

Next, we tested the hypothesis that CFL1 SUMOylation accelerates the disassembly of actin filaments using an actin depolymerization assay that monitors F-actin depolymerization through changes in the fluorescence of pyrene-labeled actin. In the absence of any CFL1 (buffer control) or in the presence of the negative control, CFL1$^{S3E}$, F-actin underwent minimal depolymerization (Fig. 4d, black and gray hollowed circles). The addition of CFL1$^{WT}$ dramatically enhanced the rate and extent of F-actin disassembly (Fig. 4d, orange hollowed circles), which were further augmented by using CFL1$^{WT}$ subjected to in vitro SUMOylation before the assay (Fig. 4d, red solid circles) or

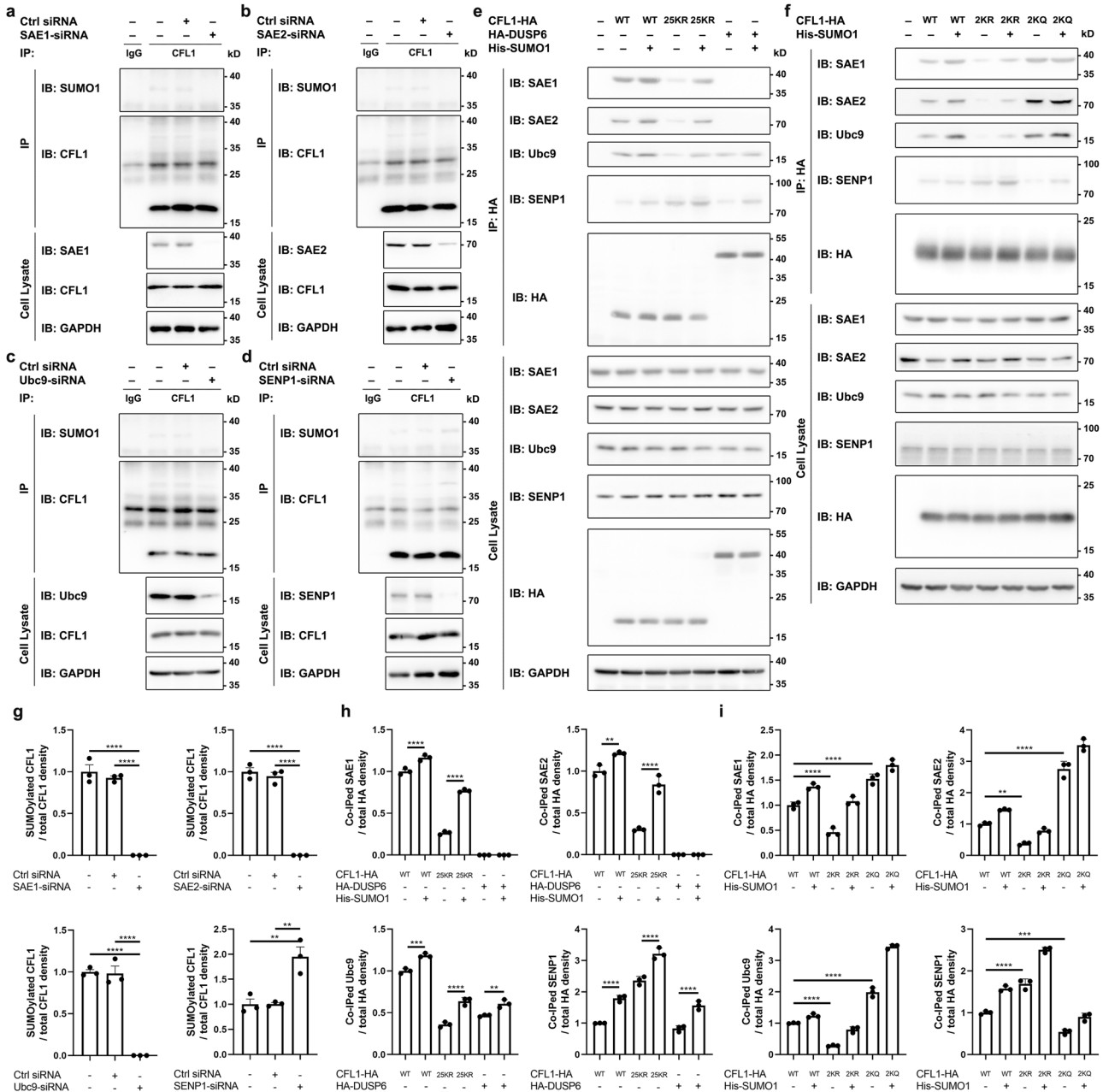

**Fig. 3 | N-α-SUMOylation of CFL1 is reversible.** The SUMOylation of CFL1 was inhibited by the knockdown of SAE1 (**a**), SAE2 (**b**), and Ubc9 (**c**), but enhanced by the knockdown of SENP1 (**d**). HEK-293T cells were transiently transfected as indicated. Cell lysates were used for De-IP by anti-CFL1 antibody followed by IB using anti-SUMO1 and anti-CFL1 antibodies. The original lysates were analyzed by IB using anti-CFL1, anti-SAE1 (**a**), anti-SAE2 (**b**), anti-Ubc9 (**c**), and anti-SENP1 (**d**) antibodies for input, as well as anti-GAPDH antibody for loading control. **e** CFL1 is physically associated with SAE1, SAE2, Ubc9 and SENP1. Lysates from HEK-293T cells transiently transfected with the vector control (−), CFL1^WT-HA, CFL1^25KR-HA, HA-DUSP6, and His-SUMO1 as indicated for 24 h were subjected to IP with anti-HA antibody, followed by IB with anti-SAE1, anti-SAE2, anti-Ubc9, anti-SENP1 and anti-HA antibodies. The original lysates were analyzed by IB using anti-HA, anti-SAE1, anti-SAE2, anti-Ubc9 and anti-SENP1 antibodies for input, and anti-GAPDH antibody for loading control. **f** Similar to (**e**) but with expression of the vector control (−), CFL1^WT-HA, CFL1^2KR-HA, CFL1^2KQ-HA, and His-SUMO1 as indicated. **g**–**i** Quantification of Western blots of (**a**)−(**f**). **g** Summary of SUMOylated CFL1 to total CFL1 presents the mean ± SEM of three independent experiments in (**a**)−(**d**); **h** Summary of coimmunoprecipitated (Co-IPed) SAE1, SAE2, Ubc9 and SENP1 normalized to total CFL1 or DUSP (HA) presents the mean ± SEM of three independent experiments in (**e**); **i** Summary of coimmunoprecipitated (Co-IPed) SAE1, SAE2, Ubc9 and SENP1 normalized to total CFL1 (HA) presents the mean ± SEM of three independent experiments in (**f**); *$p < 0.05$, **$p < 0.01$, ***$p < 0.001$, ****$p < 0.0001$, by one-way ANOVA with Tukey's multiple comparison test. All blots represent ≥3 independent experiments. The exact $p$ value and source data are provided as Source Data file.

using SUMO1-CFL1 (Fig. 4d, yellow solid circles). These results indicate that although CFL1 can induce F-actin depolymerization in the absence of SUMOylation, it is not the most efficient, and N-α-SUMOylation can accelerate the effect of CFL1 on F-actin disassembly.

As additional controls, we also tested CFL1^2KR and Me^M1+25K-CFL1 in the actin depolymerization assay. Without in vitro SUMOylation,

CFL1^2KR was similarly effective as CFL1^WT (Fig. 4e, cyan hollowed circles), suggesting that this mutation does not impair the intrinsic actin depolymerization activity of non-SUMOylated CFL1. However, this is not the case for Me^M1+25K-CFL1, which exhibited a decreased rate of inducing F-actin disassembly (Fig. 4e, blue hollowed circles). Presumably, the methylation at multiple charged sites dramatically

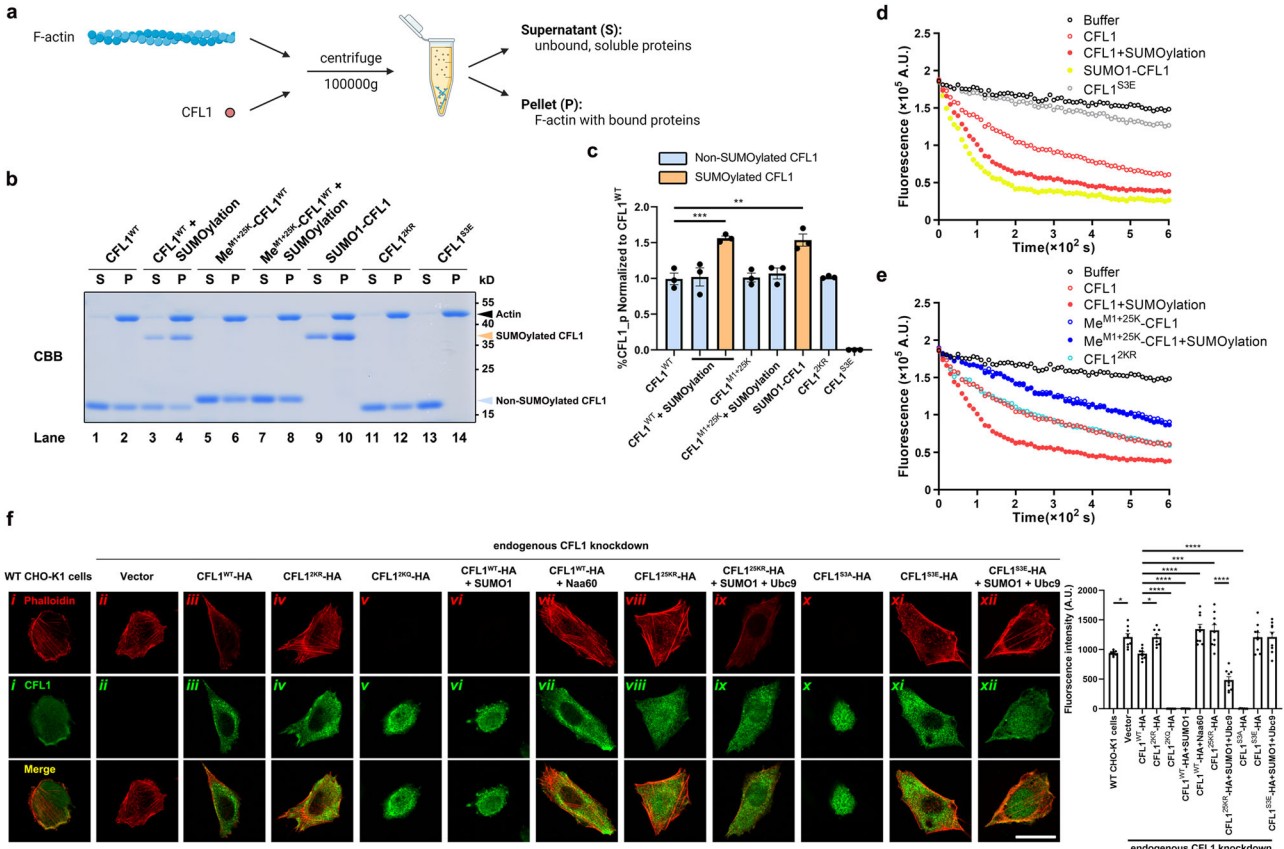

**Fig. 4 | CFL1 SUMOylation promotes F-actin depolymerization. a** Diagram for actin co-sedimentation assay. The diagram is created with BioRender.com. **b** N-α-SUMOylation of CFL1 enhances its binding to F-actin in vitro. Purified CFL1 (wild-type or mutants without or with in vitro SUMOylation as indicated) were incubated with F-actin for 30 min at room temperature. The mixtures were ultracentrifuged at 100,000 × g for 30 min. The supernatants (P) and pellets (S) were separately subjected to SDS-PAGE and stained with CBB. Orange arrowhead indicates SUMOylated CFL1; blue arrowhead indicates non-SUMOylated CFL1; black arrowhead indicates actin. **c** Quantitative analysis for (**b**). Band densities were measured separately for SUMOylated and non-SUMOylated CFL1 in the pellet (P) and supernatant (S) fractions. %CFL1_p was calculated based on the ratio of CFL1 density in P over that in P + S, which was then normalized to the value obtained for CFL1$^{WT}$ control of the same experiment. Mean ± SEM; \*\**p* < 0.01, \*\*\**p* < 0.001, compared to control by one-way ANOVA with pairwise comparison using Tukey's multiple comparisons test. **d**, **e** CFL1 SUMOylation promotes F-actin depolymerization

in vitro. Prepolymerized pyrene-labeled F-actin added to a reaction buffer alone or buffers that contained purified CFL1 (wild-type or mutants without or with in vitro SUMOylation) as indicated. Fluorescence recordings (6 min at 10-s intervals) were started immediately using a fluorescence spectrophotometer. **f** (left) CFL1 SUMOylation promotes F-actin depolymerization in vivo. CHO-K1 cells with the endogenous CFL1 stably knocked down were transfected with the indicated expression vectors. Cells were identified by immunostaining with mouse anti-CFL1 antibody followed by goat anti-mouse antibody conjugated with Alexa 488 (green). Cells were stained for F-actin using rhodamine-conjugated phalloidin (red). Scale bar = 50 μm. (right) Quantification of F-actin fluorescence intensity of (**f**). Analysis of fluorescence intensity was done at the original magnification by measuring the corrected total gray value with ImageJ software. Mean ± SEM; \*\**p* < 0.05, \*\*\**p* < 0.001, \*\*\*\**p* < 0.0001, by one-way ANOVA with Tukey's multiple comparison test. The exact *p* value and source data are provided as Source Data file.

altered the conformation of CFL1, hampering its function. Nevertheless, the lack of effect of in vitro SUMOylation on Me$^{M1+25K}$-CFL1 induced F-actin disassembly (Fig. 4e, blue solid circles) validated that the enhancing effect seen for CFL1$^{WT}$ by in vitro SUMOylation indeed resulted from the N-α-SUMOylation of this protein rather than a non-specific effect of the assay system.

The above data prompted us to examine the effect of CFL1 SUMOylation on actin polymerization status in cells. To this end, CHO-K1 cells with stable knockdown of endogenous CFL1 were transiently transfected with CFL1$^{WT}$-HA and various CFL1 mutants, alone or in combination with SUMO1, Ubc9, and Naa60 as indicated (Fig. 4f, g). Cells were stained with rhodamine-conjugated phalloidin (red) at 24 h after transfection to visualize the polymerized actin. Transfected cells were confirmed by immunofluorescence labeling with anti-CFL1 followed by Alexa 488-conjugated secondary antibody (green). As expected from the F-actin depolymerization action of CFL1[32,33], the knockdown of endogenous CFL1 resulted in an increase in F-actin fibers, and the expression of CFL1$^{WT}$ brought them back to the normal levels (Fig. 4f, i-iii). While the hypomorphic mutant, CFL1$^{2KR}$, failed to

bring down the F-actin levels, the hypermorphic one, CFL1$^{2KQ}$, led to a near complete loss of F-actin (Fig. 4f, iv-v). The effect of CFL1$^{2KQ}$ resembled that of CFL1$^{S3A}$ (Fig. 4f, x), a constitutively active mutant that can no longer be inhibited by phosphorylation, suggesting that elevating CFL1 SUMOylation in vivo dramatically enhances F-actin depolymerization. This notion is further supported by the finding from cells that coexpressed CFL1$^{WT}$ and SUMO1, which exhibited a drastic loss of F-actin as well (Fig. 4f, vi).

To confirm that the above in vivo effects on F-actin depolymerization were mediated by N-α-SUMOylation, rather than SUMO conjugation at any of the lysine residues, of CFL1, we coexpressed CFL1$^{WT}$ with Naa60. The coexpression of Naa60 prevented CFL1$^{WT}$ from bringing down F-actin levels in the CFL1-knockdown CHO-K1 cells (Fig. 4f, vii), demonstrating the importance of N-α-SUMOylation in the ability of CFL1 to mediate actin depolymerization in vivo. Secondly, although the expression of CFL1$^{25KR}$ alone was unable to decrease F-actin levels in CFL1-knockdown CHO-K1 cells (Fig. 4f, viii), boosting SUMOylation in vivo by coexpressing SUMO1 and Ubc9 allowed this hypomorphic CFL1 mutant to bring down F-actin to similar levels as

that caused by CFL1[WT] alone (Fig. 4f, ix). Given that this mutant possesses no lysine residue for internal SUMOylation, the enhancing effect of SUMO1 plus Ubc9 most likely occurred through N-α-SUMOylation of CFL1[25KR]. To rule out any nonspecific effect of SUMO1 and Ubc9 overexpression, we coexpressed them with CFL1[S3E], a nonfunctional CFL1 mutant. As expected, neither CFL1[S3E] alone nor CFL1[S3E] together with SUMO1 and Ubc9 reduced F-actin levels in the CFL1-knockdown CHO-K1 cells (Fig. 4f, xi-xii). Taken together, these results demonstrate that in cells, efficient N-α-SUMOylation is critical for CFL1 to induce F-actin depolymerization in vivo.

## Discussion

PTM by SUMO plays important roles in various cellular processes by altering the stability, conformation, interactions, and/or subcellular localization of target proteins[33]. Biochemically, SUMO shares many similarities with ubiquitin in that they are both polypeptide chains with a terminal glycine that covalently conjugates to a primary amine(s) of the substrate protein through a peptide bond. However, while ubiquitin is known to conjugate to not only the ε-NH2 groups of internal lysines but also the N-terminal α-NH2 group[23], SUMO has thus far only been studied in the context of isopeptide bond formation with the lysine ε-NH2 groups[19]. The current study, hence, provides the first evidence for the presence of N-α-SUMOylation and its importance in the substrate protein function.

Although a consensus SUMOylation site has been proposed[22], many of the recently identified SUMO conjugation sites do not adhere to this consensus[19,34], making it necessary to experimentally screen individual lysines of the substrate protein via site-directed mutagenesis[28,35]. More recently, the use of MS-based proteomics has greatly accelerated the discovery of new PTMs and their sites of action on various proteins[36]. For SUMOylation, an arginine is typically introduced to allow tryptic digest near the C-terminus of the SUMO protein so that the modified peptide from the substrate protein contains a characteristic tag, e.g., QTGG. Here, we provide multiple lines of evidence that SUMO1 is conjugated to the N-terminal α-NH2 group rather than any of the ε-NH2 groups of internal lysines of CFL1. First, the MS/MS analysis revealed the QTGG-tagged N-terminus, but not any QTGG-tagged internal lysines for endogenous CFL1, showing that the N-α-NH2 group of CFL1 can be conjugated by the engineered SUMO1 (Fig. 2b). Second, mutating internal lysines to arginines either individually or in various combinations, including the lysine-less mutant CFL1[25KR], failed to abolish CFL1 SUMOylation in cells, indicating that none of the internal lysines is essential for the conjugation (Fig. 2c and Supplementary Fig. 2b). Third, blocking the free N-α-NH2 by acetylation abolished CFL1 SUMOylation (Fig. 2d). The fact that this loss of CFL1 SUMOylation occurred in cells without changing any of the internal lysines suggests that not only is the free N-α-NH2 of CFL1 necessary but also none of the ε-NH2 groups of its internal lysines is conjugated by SUMO, at least by SUMO1 under the normal culture conditions. Finally, in in vitro SUMOylation assays, we successfully conjugated the purified recombinant CFL1 protein with SUMO1 after all its lysine ε-NH2 groups were protected by methylation, but not when the α-NH2 group of its first methionine was also protected (Fig. 2e, f), indicating that the N-α-NH2 group is both necessary and sufficient for CFL1 SUMOylation.

SUMO modification is a dynamic and reversible process that involves maturation of SUMO precursors by SENPs via their endopeptidase activity, SUMO activation by the E1 enzyme, the transfer of SUMO to the E2-conjugating enzyme Ubc9, which conjugates SUMO to the substrate protein without or with the help of an E3 ligase, and then the deconjugation of SUMO from the substrate protein by SENPs via their isopeptidase function. In the conventional SUMOylation model, only E2 and E3 enzymes need to be physically associated with the substrate protein to catalyze the conjugation and the deconjugation additionally requires the association of SENPs. Thus, it is rather surprising that CFL1 is physically associated not only with Ubc9 and SENP1

but with the heterodimeric SUMO activating E1 enzyme, SAE1/SAE2, as well (Fig. 3e, f). It is unclear if the association with SAE1/SAE2 is a special feature of N-α-SUMOylation of CFL1, but the corresponding decrease and increase in the association seen with the hypo- and hypermorphic mutants, CFL1[2KR] and CFL1[2KQ], respectively, suggest a positive correlation between the CFL1-SAE1/SAE2 physical association and CFL1 N-α-SUMOylation. The changes in their association with Ubc9 and SENP1 also suggest that these mutants alter CFL1 SUMOylation by affecting CFL1 association with the conjugating and deconjugating enzymes in opposite ways. Moreover, that overexpression of SENP1 abolished CFL1 SUMOylation (Fig. 1a, b) supports the reversibility of SUMO conjugation at the N-NH2 group and the involvement of endopeptidase activity of SENP1 in the deconjugation. Furthermore, like in many cases of conventional SUMOylation at internal lysine ε-NH2 groups[19], the SUMO modified CFL1 accounts for a very small fraction of the total CFL1 proteins, consistent with the N-α-SUMOylation also being highly dynamic and reversible. How N-α-SUMOylation is regulated, whether it is common or rather restricted to few proteins, does it occur simultaneously on the same protein with conventional SUMOylation on internal lysine ε-NH2 groups, and in which way it is processed differently from the conventional SUMOylation are questions that warrants future investigation.

PTMs play important roles in regulating CFL1 function by allowing local control for enhanced versatility. Consistent with the previous finding that self-regulating cofilin-mediated actin dynamics can drive motility without posttranslational regulation[18] in in vitro assays, all recombinant CFL1 proteins, which represent non-phosphorylated forms, except for the phosphomimetic mutant CFL1[S3E], were capable of binding to F-actin and inducing its depolymerization (Fig. 4b–e). These included the SUMOylation hypomorphic mutant, CFL1[2KR], which exhibited similar F-actin binding and disassembly as CFL1[WT] in the absence of SUMOylation. However, in CFL1-knockdown CHO-K1 cells, the expression of CFL1[2KR] did not bring down the over accumulation of F-actin as that of CFL1[WT] (Fig. 4f). On one hand, the spontaneous SUMOylation of CFL1[WT] is higher than that of CFL1[2KR] when expressed in cells (Supplementary Fig. 3a, b), which should result in more efficient F-actin binding and F-actin disassembly for CFL1[WT] than for the 2KR mutant independently of CFL1 phosphorylation, as shown in the in vitro assays (Fig. 4b–e). On the other hand, phosphorylation at Ser3 is the best known negative PTM on CFL1 function[6]. Different from the in vitro assays, CFL1 proteins in live cells are mostly phosphorylated to keep them from spontaneously active. Interestingly, increasing SUMOylation in cells suppressed CFL1 phosphorylation at Ser3 (Supplementary Fig. 7a), implicating the additional presence of a phosphorylation-dependent mechanism modulated by the N-α-SUMOylation in the cellular context to further magnify CFL1 activation. Plausibly, the attachment of the large SUMO group at the N-terminus of CFL1 blocks the access of LIM kinases to Ser3 to catalyze the phosphorylation, reducing the available phosphorylation substrate of CFL1. Therefore, in cells, N-α-SUMOylation of CFL1 engages both phosphorylation-dependent and -independent mechanisms to augment its effects such that despite the low proportion of the N-α-NH2 SUMO-modified fractions, this PTM exerts a profound effect on the ability of CFL1 to depolymerize F-actin (Supplementary Fig. 7b). This may explain the nearly complete lack of effect of CFL1[2KR] on F-actin disassembly when expressed in the CFL1 knockdown CHO-K1 cells, even though this SUMOylation hypomorphic mutant is able to bind and depolymerize F-actin in in vitro assays. It also explains why increasing CFL1 SUMOylation, either by using a hypermorphic mutation or co-expression of SUMO1 with CFL1[WT], enhanced F-actin disassembly so strongly such that the cells even round up, as in the case of expressing the constitutively active mutant, CFL1[S3A] (Fig. 4f).

Notably, some caution should be taken when considering the effect of SUMOylation on CFL1 regulation of actin dynamics as both the in vitro actin depolymerization assay and phalloidin staining suffer

some drawbacks that could affect the data interpretation. For example, ADF/cofilin binding is known to quench the fluorescence of pyrene[37,38], which could contribute to the fluorescence decrease during the early phase (likely the first minute) of the actin depolymerization assay. Second, because cofilin binding to F-actin interferes with phalloidin binding[39], the decreased phalloidin staining in the cell could also implicate a more stable interaction of SUMOylated CFL1 with F-actin. These possibilities remain to be evaluated. Third, the use of paraformaldehyde fixation and Triton X-100 permeabilization in the in vivo F-actin staining assay can disrupt cofilin-actin bundle staining[40]. As a result, relying solely on phalloidin staining may not offer an optimal approach for assessing polymerized actin. To delve deeper into the dynamic regulation of F-actin by CFL1 SUMOylation, we intend to explore alternative and more effective methodologies. Fourth, cofilin is equipped with a bipartite nuclear localization sequence (NLS) and plays a pivotal role in the transportation of actin into the cell nucleus[41,42]. Given this premise, it is reasonable to hypothesize that CFL1 could undergo SUMOylation by SUMOs, which are primarily recognized as nuclear proteins. This could subsequently lead to the regulation of F-actin depolymerization. Moreover, many studies have demonstrated that other actin-binding proteins, such as actin-interacting protein 1 (AIP1), cyclase associated protein (CAP), coronin, tropomyosins (TPM), cortactin and actin-related proteins-2/3 (Arp2/3), can potently modulate ADF/cofilin's ability to act on the actin cytoskeleton[43]. We have not yet explored the impact of CFL1 SUMOylation on these actin-binding proteins, which requires further research. Finally, it's worth noting that our partial findings rely on overexpression experiments, which may not completely eliminate the potential for artifacts. Moving forward, it becomes imperative to establish a biologically authentic model that can provide more robust insights into the precise functions of N-terminal SUMOylation on CFL1.

In conclusion, our data reveal a previously unknown form of SUMO conjugation at the N-terminal α-amino group of the substrate protein. We show that CFL1 is SUMOylated at N-$\alpha$-NH$_2$ but not $\varepsilon$-NH$_2$ group of any of the internal lysine residues, although some of the lysine residues of CFL1 may indirectly affect its N-$\alpha$-SUMOylation through an allosteric effect on the physical association of CFL1 with SUMO conjugation and deconjugation machineries, including E1 and E2 enzymes and SENP1. Functionally, the N-$\alpha$-SUMOylation facilitates CFL1 binding to F-actin and F-actin depolymerization via both CFL1 phosphorylation-dependent and -independent mechanisms. We conclude that the N-$\alpha$-SUMOylation of CFL1 is critically involved in its regulation of actin cytoskeleton dynamics, providing added versatility to cellular responses to physiological stimuli and stress.

## Methods

### Ethical statement

The research presented in this study adheres to the ethical regulations at Shanghai Jiao Tong University School of Medicine. All protocols involving animal experiments were conducted in accordance with the guidelines for the Care and Use of Laboratory Animals of Shanghai Jiao Tong University School of Medicine and were approved by the Institutional Animal Care and Use Committee (IACUC).

### Antibodies and reagents

Anti-HA (H6908, 1:3000), anti-Flag agarose (A2220) and N-ethylmaleimide (NEM; E3876, 20 mM) were purchased from Sigma-Aldrich. Anti-SUMO1 (4940, 1:1000), anti-SAE1 (13585, 1:1000), anti-SAE2 (8688,1:1000) and anti-GAPDH (2118, 1:5000) were purchased from Cell Signaling Technology. Mouse anti-CFL1 (66057-1-Ig, 1:1000), rabbit anti-CFL1 (10960-1-AP, 1:1000), anti-GST (66001-2-Ig, 1:3000) and anti-His (66005-1-Ig, 1:1000) were purchased from Proteintech. Anti-Ubc9 (ab33044, 1:1000) and anti-SENP1 (ab108981, 1:1000) were purchased from Abcam. Anti-HA agarose (26181) and protein A/G

agarose (20422) were purchased from Pierce. Protease inhibitor cocktail (B14001) was purchased from Bimake.

### Experimental animals

Mice were housed in a controlled environment with a 12-h dark/light cycle and were provided with ad libitum access to both food and water. The mice utilized in the study were males and were employed at 8 weeks of age. The specific mouse strains employed were maintained on the C57BL/6 background.

### Plasmids and RNA interference

Mouse CFL1 cDNA was amplified by PCR from mouse brain tissues and inserted into the pCMV-HA-C (Clontech) vector to obtain CFL1-HA. Various CFL1-HA mutations (K13R, K19R, K22R, K30R, K31R, K33R, K34R, K44R, K45R, K53R, K73R, K78R, K92R, K95R, K96R, K112R, K114R, K121R, K125R, K126R, K127R, K132R, K112/114R, K112/114Q and 25KR) were generated using the KOD Plus mutagenesis kit (TOYOBO). Myc-Naa60 was purchased from SinoBiological (MG52477-CM). His-SUMO1, Flag-SUMO1, RGS-SENP1, RGS-SENP$^{CS}$, Flag-PIAS1, Flag-PIAS2α, Flag-PIAS2β, Flag-PIAS3 and Flag-PIAS4 plasmids were previously described[36,44].

SAE1, SAE2, Ubc9 and SENP1 siRNA duplexes were synthesized by Tsingke. The siRNA oligos specific for SAE1 (5'-AGCGAGCUCAGAAU-CUCAA-3') and the corresponding scrambled siRNA oligonucleotide (5'-UUCUCCGAACGUGUCACGU-3'), siRNA oligos specific for SAE2 (5'-GCCCGAAACCAUGUUAAUAGA-3') and the corresponding scrambled siRNA oligonucleotide (5'-GCCUAACTGTGTCAGAAGGAA-3'), siRNA oligos specific for Ubc9 (5'-GGCCAGCCAUCACAAUCAATT-3') and the corresponding scrambled siRNA oligonucleotide (5'-GAGCCTA-CAACCCATGACTTA-3'), or siRNA oligos specific for SENP1 (5'-UCCUUUACACCUGUCUCGAUGUCUU-3') and the corresponding scrambled siRNA oligonucleotide (5'-CUUCCUCUCUUUCUCUCCCUU-GUGA-3') were transfected at a final concentration of 100 nM into HEK-293T cells using Lipofectamine 3000 (L3000015, Invitrogen) according to the manufacturer's protocol.

### Cell culture and transfection

CHO-K1 cells (GNHa 7), HEK-293T cells (GNHu17) and Neuro-2a (TCM29) were purchased from Cell Bank/Stem Cell Bank, Chinese Academy of Sciences (Shanghai, China). All types of cells were cultured at 37 °C in 5% CO$_2$ humidified incubator. CHO-K1 cells were grown in Ham's F12 medium containing 10% fetal bovine serum (FBS, Invitrogen). HEK-293T cells and Neuro-2a cells were grown in DMEM (high glucose) supplemented with 10% FBS. For transfection (all cell types), cells grown into -80% confluence were transfected with the desired plasmids together using Lipofectamine 3000 (L3000015, Invitrogen) or 1 mg/ml polyethylenimine (PEI, 23966, Polysciences) according to manufacturers' instructions.

### Immunoprecipitation and Western blotting

Immunoprecipitation (IP) and Western blotting (IB) were performed as described previously with minor modifications[29]. The non-denaturing IP and denaturing IP (De-IP) differed in the lysis buffer and the procedure of making cell lysates. For non-denaturing, cells were lysed with the Triton X-100 lysis buffer (20 mM NEM, 20 mM Tris-HCl, 150 mM NaCl, 1% Triton X-100, 2 mM EDTA, 10% glycerol, pH 7.5) containing protease inhibitor cocktail on a rotator at 4 °C for 1 h. For De-IP, cells and brain tissues (extracted from 8-week-old male mice) were lysed in SDS lysis buffer (50 mM Tris-HCl, 3% SDS, 5% glycerol, pH 7.5). The lysates were then boiled for 10 min at 95 °C before being diluted 10-fold using NP-40 lysis buffer (20 mM NEM, 50 mM Tris-HCl, pH 7.4, 150 mM NaCl and 1% Nonidet P-40) containing protease inhibitor cocktail. The subsequent steps were identical for both methods. Briefly, after a centrifugation at 21,130 × $g$ at 4 °C for 20 min, the supernatant was collected and incubated with the desired primary

antibody at 4 °C overnight. Protein A/G Sepharose beads were added to purify the proteins. After incubation at 4 °C for 3 h, the beads were washed using Triton X-100 lysis buffer for three times. The beads were boiled in SDS sample buffer and then subjected to SDS-polyacrylamide gel electrophoresis (PAGE) and IB. Gray values of IB bands were quantified using ImageJ software (National Institutes of Health).

## Mass spectrometry

For SUMOylation site identification of endogenous CFL1, Neuro-2a cells were transfected with His-SUMO1$^{E93R}$. For N-terminal acetylation identification of CFL1, Neuro-2a were transfected with Myc-Naa60. At 24 h post transfection, cells were lysed with the SDS lysis buffer followed by diluting in the NP-40 lysis buffer containing protease inhibitor cocktail as described above for De-IP. After a centrifugation at 21,130 × g at 4 °C for 20 min, the supernatant was collected and incubated with anti-CFL1 antibody at 4 °C overnight before protein A/G beads were added. After 3 h incubation at 4 °C, immunoprecipitants were washed at 4 °C sequentially with Triton X-100 lysis buffer (see above) for four times, followed by 5 mM NH$_4$HCO$_3$ twice. Proteins were eluted from the beads with 0.15% trifluoroacetic acid (T6508, Sigma-Aldrich). After centrifugation at 4 °C for 15 min at 15,870 × g, the supernatant was collected and dried in vacuum. The dried samples were redissolved in 25 mM NH$_4$HCO$_3$ followed by digestion with trypsin (Promega) at 37 °C overnight and was dried before LC-MS/MS analysis.

For LC-MS/MS, the samples were resuspended with 20 ml Buffer A (water with 0.1% formic acid) and analyzed by online nanospray LC-MS/MS on a Q Exactive HF (Thermo Fisher Scientific) coupled to an Acquity UPLC M-class (Waters Corporation). Three microliters of peptides were loaded (analytical column: Waters nanoEase M/Z HSS C18 T3, 75 μm × 25 cm) and separated with a 60 min linear gradient, from 4 to 30% Buffer B (acetonitrile with 0.1% formic acid). The column flow rate was maintained at 500 nl/min with the column temperature of 40 °C. The electrospray voltage of 2 kV versus the inlet of the mass spectrometer was used. The mass spectrometer was run under data dependent acquisition mode and automatically switched between MS and MS/MS mode. The parameters were: (1) MS: scan range (m/z) = 350–1600; resolution = 60,000; AGC target = 3e6; maximum injection time = 50 ms; include charge states = 2–7; (2) HCD-MS/MS: resolution = 15,000; isolation window = 1.6; AGC target = 1e5; maximum injection time = 100 ms; collision energy = 30.

Tandem mass spectra were processed by PEAKS Studio version X (Bioinformatics Solutions Inc). PEAKS DB was set up to search the uniprot_proteome_mus_musculus_201907 database (ver 201907, 22290 entries) assuming trypsin as the digestion enzyme. PEAKS DB was searched with a fragment ion mass tolerance of 0.02 Da and a parent ion tolerance of 7 ppm. Acetylation (Protein N-term) and QTGG modification were specified as the variable modifications. The peptides with −10lgP ≥ 20 and the proteins with −10lgP ≥ 20 and containing at least one unique peptide were filtered.

## Cloning, mutagenesis, and expression of recombinant proteins

The full-length mouse CFL1 cDNA was subcloned in E. coli expression vector pE-SUMO3. All mutants were constructed by PCR mutagenesis using the KOD Plus mutagenesis kit (TOYOBO). The expression plasmid construct was transformed into BL21 (DE3) cells and grown in 2× YT medium at 37 °C until optical density at 600 nm reached 0.8. Then isopropyl-β-D-1-thiogalactopyranoside (IPTG) was added to a final concentration of 0.2 mM, and the culture was transferred to a 25 °C shaker and further incubated for 12 h. The cells were harvested by centrifugation and resuspended in ice-cold buffer (20 mM Tris-HCl, pH 7.4, 0.5 M NaCl, and 20 mM imidazole) and lysed by a high-pressure cell cracker. After centrifugation at 44,720 × g at 4 °C for 20 min, the supernatant was collected and loaded onto a His column and further

purified by a Q column. The SUMO3 tag was removed by protease SENP2 and the samples were finally purified by a His column.

## In vitro methylation

The CFL1 protein and SUMO3-fused CFL1 protein were dialyzed in the dialysis buffer (50 mM HEPES, pH 7.5, 250 mM NaCl). Then, 20 μl of 1 M dimethylamine-borane complex and 40 μl of 1 M formaldehyde were added per ml of the protein solution. The solution was gently mixed and incubated at 4 °C. After 2 h, another 20 μl of 1 M dimethylamine-borane complex and 40 μl of 1 M formaldehyde were added and the incubation continued for 2 h. Finally, 10 μl of 1 M dimethylamine-borane complex were added and the reaction was incubated overnight at 4 °C. The methylation reaction was terminated by dialysis and methylation was confirmed by LC-MS (see above).

## In vitro SUMOylation assay

In vitro SUMOylation assay was performed according to manufacturer's instructions using an assay kit (BML-UW8955-0001, Enzo Life Sciences). Briefly, 2 μg of the CFL1 protein (CFL1$^{WT}$, Me$^{M1+25K}$-CFL1 and Me$^{25K}$-CFL1) was incubated in SUMOylation buffer with a reaction mixture containing recombinant E1 enzyme, E2 enzyme, and SUMO1 protein in the presence or absence of ATP for 2 h at 37 °C. For SDS-PAGE, the reaction was terminated by diluting in SDS sample buffer. For F-actin co-sedimentation assay, the solution was store at −80 °C and incubated with F-actin as described below.

## Proximity ligation assay (PLA)

Proximity ligation assay (PLA) was conducted following the manufacturer's guidelines using an assay kit (DUO92101, Sigma-Aldrich). CHO-K1 cells were fixed with 4% paraformaldehyde for 15 min at room temperature. Subsequently, the cells were subjected to primary antibody incubation, followed by incubation with PLA probes (anti-mouse and anti-rabbit IgG antibodies conjugated with oligonucleotides). Ligation and amplification steps were carried out according to the manufacturer's instructions. Confocal images were acquired using an Olympus LSM800 confocal microscope equipped with a 63× oil immersion objective lens. The quantification was performed using ImageJ software from the National Institutes of Health.

## F-actin co-sedimentation assay

F-actin co-sedimentation assay was performed as described previously[45]. Briefly, 2 mg/ml actin from rabbit muscle (A001041, Sangon Biotech) was polymerized for 60 min at room temperature in 10× polymerization buffer (0.2 M Tris-HCl, pH 7.5, 1 M KCl, 20 mM MgCl$_2$, and 1 mM dithiothreitol). Purified CFL1 protein (5 μg of wild type or mutants) was incubated with 10 μg of F-actin in 200 μl of 10 mM Tris-HCl (pH 8.0) for 30 min at room temperature. Then, the mixtures were ultracentrifuged at 100,000 × g at 4 °C for 30 min. The supernatants and pellets were separately subjected to SDS-PAGE and stained with Coomassie brilliant blue (CBB). Quantification was performed by ImageJ software (National Institutes of Health).

## Actin depolymerization assay

Actin depolymerization activities were measured by monitoring the changes in fluorescence intensity of pyrene-labeled actin, as described previously[46]. Pyrene-labeled monomeric G-actin (CS-AP07, Cytoskeleton) was dissolved in G-actin buffer (5 mM Tris-HCl, pH 8.0, 0.2 mM CaCl$_2$, 0.5 mM dithiothreitol, and 0.2 mM ATP) and then polymerized for 60 min at room temperature in 10× polymerization buffer (see above). Pyrene-F-actin (5 μM) was mixed with purified CFL1 proteins, wild type and mutants (5 μM). Fluorescence levels were immediately recorded at 10-s intervals for 6 min by a fluorescence spectrophotometer. The excitation and emission wavelengths were 350 and 407 nm, respectively.

## Immunofluorescence staining

Twenty-four hour after transfection, CHO-K1 cells were fixed in 4% paraformaldehyde for 15 min, permeabilized in 0.1% Triton X-100, and blocked with 5% goat serum for 30 min at room temperature. Cells were stained for the presence of CFL1 with mouse anti-CFL1 antibody (1:500) and goat anti-mouse IgG conjugated with Alexa 488 (green; R37120, 1:500, Invitrogen). Filamentous actin was visualized by staining the cells with rhodamine-conjugated phalloidin (red; 1:50) according to manufacturer's instructions. Confocal images were captured on a Leica SP8 confocal microscope (Leica) using a 63× oil immersion objective lens. Quantification was performed by ImageJ software (National Institutes of Health).

## Statistics and reproducibility

Summary data are presented as mean ± SEM with statistical significance assessed by Student's $t$ test for two-group comparison or one-way analysis of variance (ANOVA) for more than two groups. $p < 0.05$ is considered statistically significant. One representative biological replicate of an experiment is presented in the figures. All experiments were performed three or more times independently under identical or similar conditions, except when indicated in the figure legends.

## Reporting summary

Further information on research design is available in the Nature Portfolio Reporting Summary linked to this article.

## Data availability

Mass spectrometry data that support the findings of this study have been deposited in PRIDE with the accession code PXD038392. The data generated in this study are provided in the Supplementary Information and Source Data file. Source data are provided with this paper.

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

## Acknowledgements

This study was supported by grants from the National Natural Science Foundation of China (31830031, 82071510, 31761163002 and 31671053 to Y.L.), the National Key Research and Development Program of China (2020YFA0803602 to Y.L.), and the Innovative Research Team of High-Level Local Universities in Shanghai (SHSMU-ZDCX20211102).

## Author contributions

Y.L. conceived the studies and designed the experiments. W.W. and Q.D. carried out the biochemical and immunofluorescence experiments. Y.Y., X.G. and Q.Z. prepared recombinant proteins. J.Z. and O.H. assisted with data analysis and analyzed results. W.W., J.F., O.H., J.C. and M.X.Z. discussed the results. J.F., O.H., J.C. and M.X.Z. provided technical support. W.W., M.X.Z. and Y.L. wrote the manuscript with inputs from the other authors. All authors discussed and commented on the manuscript.

## Competing interests

The authors declare no competing interests.
