## [Peer Review File · Nature Communications]

Reviewers' Comments:

Reviewer #1:

Remarks to the Author:

Summary of Critique: Cofilin-1, the ubiquitously expressed mammalian isoform, is here shown to be a substrate for SUMOylation, which would not in and of itself be surprising since cofilin has been characterized as a substrate for posttranslational modification (PTM) by ubiquitin and another ubiquitin-like modifier, NEDD-8. However, for these other cofilin PTMs, as well as all other known proteins that are substrates for SUMOylation, the modification occurs on epsilon-amino groups of lysine residues, whereas for cofilin the SUMO is on the N-terminal alpha-amino group. This reversible modification has implications for the phosphorylation of cofilin on its Ser3 residue, which is inhibitory to its interactions with both G- and F-actin. The authors go to great lengths using multiple methods to convincingly demonstrate this point. It is an unexpected finding because most cofilin in cells is N-acetylated on the penultimate alanine following its demethionylation, and, as noted herein, N-acetylation is an irreversible modification, suggesting that SUMOylation occurs on newly synthesized cofilin. Unfortunately, this aspect was not addressed through cell biological studies. What was shown is that SUMOylated cofilin binds somewhat better to F-actin than WT cofilin using in vitro co-sedimentation assays. The actual studies of SUMO-cofilin function in a cell biological context are the weakest part of this manuscript and rely on some "standard" methods that have serious interpretation problems when performed with cofilin-actin in complex (addressed below).

Specific Points.

The intensity of the SUMOylated cofilin band on gels is very weak in Figure 1a,b and even in the prolonged exposure in Extended Figure 1. The authors do not address this until the discussion where they mention the small amount of SUMO-cofilin in explaining its oversized effect on actin dynamics suggesting its very rapid turnover, similar to what has been reported for NEDDylated cofilin on lysine 112. The authors should address this point up front when the blots are first presented so that the reader does not think small amounts of a modification might be insignificant in terms of functional consequences. It would be worthwhile to present the actual ratio of putative SUMO-cofilin to WT-cofilin in immunoblots of extracts from cells/tissues in Figure 3, but visualized using the same cofilin antibody, even though the band for SUMO-cofilin likely contains other mono-ubiquitinated, NEDDylated and SUMOylated species. Non-cofilin aficionados would probably benefit from seeing such a quantitative comparison to understand how small changes can have big impacts.

Blots are presented with all of the appropriate controls but there is no quantification shown for any of the changes that seem to be significant. Since blots were repeated at least 3 times, there should be enough data available to provide graphs from which significance in the changes in SUMO-cofilin can be determined for Fig 1 a,b,c; Fig 2 d,f; and Fig 3.

Also, in Figure 3b, why does control siRNA cause the seemingly large increase in SAE2 on the blot? This increase is certainly as significant as most other changes that are discussed.

With some quantifiable data included in the immunoblot figures, the biochemical and structural data presented in this manuscript all appear to be quite adequate and leave little room to doubt the conclusions that cofilin is SUMOylated on its N-terminus.

However, the functional significance of this finding with respect to how it affects in vitro actin dynamics and cell biological function needs to be better addressed.

Functional studies are all summarized in Figure 4. Co-sedimentation analyses with muscle F-actin strongly suggests that SUMO-cofilin binds better than WT to F-actin and thus appears in the pellet fraction in a higher ratio compared to WT cofilin. If these sedimentation studies were performed in the TLA-100 bench-top centrifuge in tubes holding 200 ul (as suggested but not specified in the methods), the authors should use the K value of the rotor and the speed to determine what size filaments might still be in solution after 30 minutes ($t=K/S$) where t is time, K is the rotor clearance value (usually obtained from the rotor booklet of the manufacturer) and S is the Svedberg units for the proteins in question. This was done for actin and ADF/cofilins in 2004 (doi: 10.1021/bi049797n) and it is surprising that even at 436,000xg (max speed used in this publication) it takes longer than 30 min to clear small pieces of cofilin-actin. At only 100,000xg as reportedly used here, larger fragments will still be in solution. This is an important point because of the cooperative binding of cofilin along a filament. Thus, regions of cofilin-saturated and "naked" F-actin occur together on a single filament, with

severing often occurring at the junction between them. Is SUMO-cofilin still able to bind to F-actin in a cooperative manner or is there any steric effect of the SUMO on its binding to adjacent actins?

One commonly used *in vitro* study that could go a long way toward answering some of these questions and is missing here is a study fluorescently tagged F-actin anchored on the surface of a coverslip while allowing various forms of cofilin to flow into the system. Both severing and end depolymerization could then be compared between SUMO- and WT cofilin.

The studies on depolymerization of pyrene-F-actin also require careful interpretation. Cofilin binding to pyrene-F-actin causes quenching of the pyrene fluorescence which can be misconstrued as depolymerization. Thus, a second method to confirm the depolymerization is required. The enhanced binding of SUMO-cofilin to F-actin shown in Fig 4 C might account for its apparent "stimulation of depolymerization" in Fig 4D, but may actually arise from quenching of fluorescence, although if SUMO-cofilin still binds cooperatively, the time course of 3 min for complete loss of pyrene signal might not fit with the rate of binding. The authors need to examine the rates for both binding and fluorescence quenching from previous publications on WT cofilin.

Finally, the data in Figure 4F assumes that fluorescent phalloidin binds to all cellular filamentous actin and thus its decreased degree of labeling shows loss of F-actin. However, phalloidin does not bind to cofilin-actin. Furthermore, using Triton as the cell permeabilization agent prevents cofilin immunolabeling of cofilin-actin filaments. Together, these events have led to a decade of misinterpretation of decreased phalloidin binding as meaning less assembled actin filaments. This information concerning cofilin-actin been in the literature for many years but is widely ignored since Triton permeabilization is the "go to" method in virtually every immunolabeling textbook. To visualize cofilin-actin filaments by cofilin immunolabeling requires permeabilization using either 100% -20o C methanol or acetone. Acetone permeabilization allows some phalloidin staining but even so, phalloidin does not stain cofilin-actin filaments. A recent paper demonstrating this very well in the proximal filopodia of neuronal growth cones was published by Hylton et al. (<https://doi.org/10.1038/s41467-022-30116-x>) with a follow up methods paper (DOI: 10.1007/978-1-0716-2811-9_18). Thus, the loss of phalloidin staining could arise through enhanced SUMOcofilin-stabilized F-actin which competes with myosin II for actin binding and thus reduces tension in cells through actin cables; competition between cofilin and myoII binding regulates cortical and nuclear tension in normal cell behavior. It could also be due to very specific effects on focal adhesions- suggested by the cell rounding. These adhesions may be sites of local cofilin synthesis which could be the major targets for SUMOylation of newly synthesized cofilin and worth looking at in detail. Given this information, the data in Fig 4F is really not very useful without additional panels demonstrating that there is not and increase of cofilin-actin filaments. Staining focal adhesions in spread cells and observing their change during SUMOylation of cofilin would also be a worthwhile addition. Cell rounding might result from a direct effect of SUMO on actin depolymerization and not adhesion, but this is hard to address without being able to acutely turn on SUMO, something that could be done in future studies. Avoid studying cofilin-actin filaments *in vivo* using Lifeact or the other F-actin binding peptide chimeras with fluorescent proteins- these do not visualize cofilin-actin filaments because they recognize the same filament conformation as phalloidin.

The authors are certainly aware that cofilin disassembly of F-actin *in vivo* involves the interplay of many other proteins, some of which could be strongly affected by SUMO-cofilin. Among these are AIP1 (aka WDR1), which enhances severing of cofilin-actin, and essential cyclase-associated proteins (CAP1/2) which binds cofilin-actin filaments releasing cofilin and enhancing nucleotide exchange on the ADP-actin (to ATP-actin) as it is released. While studying how SUMO-cofilin affects the function of AIP1 or CAPs is not an expectation for this manuscript, acknowledging the broader aspects of SUMO-cofilin involving other proteins would be a useful addition to the discussion.

Reviewer #2:

Remarks to the Author:

In this manuscript, the authors report on their discovery of N-terminal conjugation of a substrate protein (CFL1) with the SUMO1 peptide. First, they show that CFL1 is modified by SUMO1 *ex vivo* in cultured cells, *in vitro* in a reconstituted sumoylation system, and also *in vivo* in mouse brains.

Sumoylation *ex vivo* seems to be reversible upon overexpression of the SUMO protease SENP1, but not of a catalytically inert SENP1 mutant. Then, using multiple complementary approaches, they demonstrate that this modification occurs on the N-terminal amino group of CFL1, rather than on any of the internal lysines, which is indeed a novel and interesting finding.

To identify the major SUMO1 conjugation site on CFL1, the authors mutate each of the 25 lysine residues on CFL1 into arginines, and also create a 25KR mutant where all 25 lysines are mutated. Even though the K112R and K114R mutants exhibit loss of sumoylation to a certain degree, none of the mutants, including the 25KR mutant display total loss of sumoylation, suggesting that CFL1 could be modified outside one of these lysines.

They perform mass spectrometry analyses on endogenous CFL1 (pulled-down from Neuro2A cells) by taking advantage of an overexpressed His-SUMO1E93R mutant. MS/MS data revealed that the major conjugation site carrying the QTTG signature following tryptic cleavage was the N-terminal alpha amino group. Protection of this group by acetylation (upon Naa60 overexpression) abolished CFL1 sumoylation. They also perform methylation assays, to "methyl-protect" or "methyl-block" all of the internal lysine epsilon amino groups, or those plus the N-terminal alpha amino group, then perform *in vitro* sumoylation. These experiments also validated that the free internal lysine side chains were dispensible for SUMO1 conjugation, but not the amino end.

Finally, the authors perform several functional experiments to demonstrate that sumoylated CFL1 (either *in vitro* sumoylated CFL1, or an engineered version of CFL1 with an N-terminal SUMO1 tag that mimics its conjugation) binds more efficiently to F-actin compared to non-sumoylated CFL1 and induces depolymerization more efficiently.

Overall, these are novel and interesting findings. Firstly, a growing number of proteins now appear to be sumoylated outside of the traditional sumoylation consensus motif ψ -K-X-D/E, ^[15] and these non-traditional conjugation sites are sometimes difficult to identify. In such cases where finding the major SUMO conjugation site proves to be difficult, this manuscript now suggests that researchers in the field should also consider the possibility that a substrate may be modified at the N-terminus. Conjugation of ubiquitin to alpha amino groups has been reported, but this was previously unrecognized for the SUMO field. Secondly, although CFL1 is known to be regulated by multiple PTMs, this manuscript now unveils another layer of regulation for this important substrate.

In conclusion, I recommend this well-written and nicely organized manuscript be published provided that the authors also address the following comments:

1) It is rather confusing that the authors use different cell lines throughout the manuscript. For example, some of the cell-based experiments in Fig 1 were performed in CHO-K1 cells. Then, the mass spec results on endogenous CFL1 were obtained using the Neuro2A cell line. In Figure 3, however, SUMO E1, E2 enzyme knock-downs or the experiments probing physical interactions between CFL1 and E1 subunits (or E2) were all performed in HEK293 cells. Why is that? How do the authors justify this non-uniformity? It is certainly OK and even advisable to repeat an experiment and validate its result in a second cell system; however, employing different cell lines for different experiments should be avoided, or at least justified. Also, is there a particular reason why the authors use mouse cerebral cortexes for the experiment in Fig 1D?

2) It would be interesting to determine if CFL1 also undergoes SUMO2/3 conjugation, and whether that also occurs at the N-terminal alpha amino group. It should be relatively easy to repeat some of the simple experiments (i.e. Fig 1A, 2C, 2D, 2G) using His-SUMO2/3.

3) In all IP (immunoprecipitation) experiments, starting from Fig 1A, the band corresponding to sumoylated CFL1 is extremely weak and barely visible. Indeed, for most protein substrates, only a small portion of the entire pool gets modified and the authors discuss this issue. However, this seems to be extreme for CFL1. Because the authors are using His-tagged SUMO1 in some cases (i.e. Fig 1A), it would be nice to perform an "anti-His" Western blot on their immunoprecipitated samples. One would wonder if that yields a better signal.

4) In Fig 3A, the IP seems to have been performed on endogenous proteins (endogenous CFL1 and

endogenous SUMO1). This experiment, compared to the experiment in Fig 1A where SUMO1 is overexpressed, yields a sumoylated CFL1 band that is of the same weak intensity. Shouldn't the authors detect enhanced sumoylation of CFL1 when they overexpress SUMO1, as in Fig 1A? Does that happen only in the presence of UBC9?

5) Figure 3D: in the text, the authors say "knockdown of SENP1 markedly increases the SUMOylation levels of CFL1". The results shown in this Western blot are not strong enough to support this conclusion. I can barely see a band, let alone a remarkable increase in the band's intensity. How many times has this experiment been performed? Is this the most representative WB? Data here do not support the conclusion.

6) SENP1 is a predominantly nuclear protease. How do the authors explain its "desumoylase" effect on CFL1, a cytoplasmic protein?

7) The physical association between CFL1 and the E1 enzyme subunits is interesting and confusing at the same time, and deserves further exploration and validation. The authors should make an effort to prove this potential physical interaction using a complementary approach, for example, proximity ligation assays (PLA). If the authors end up employing PLA, they should also try to detect CFL1 / SENP1 interactions.

8) In Figure 4F, the authors use CHO-K1 cells where the endogenous CFL1 is stably knocked down. The labeling on the first panel (far left in the figure) "without CFL1 KD" is confusing. They should just write "cells without CFL1" or "CFL1 KD cells".

9) Again, in Figure 4F, it is difficult, if not impossible, to see a difference in the phalloidin signals between i (or ii) and iii, contrary to what the authors claim. Is it possible to quantify these signals? The authors should use a software to quantify signals from at least 20-30 cells for each experimental group, and represent the results on a graph (along with the averages and error bars). Without quantifications, it is hard to make comparisons and draw conclusions. For example, why is it that CFL25KR (viii), which is sumoylated as well as CFL-WT (iii), seems to be less effective? Why does it need UBC9 and SUMO1 co-expression (ix) to be only as effective as CFL-WT without UBC9/SUMO1?

10) Again, in Figure 4F, the authors can add a panel where they treat cells with a sumoylation inhibitor (i.e. ML792).

Reviewer #3:

Remarks to the Author:

Ubiquitin is predominately linked to lysines in substrates, but can also be linked via amino termini of proteins. Whether other ubiquitin-like proteins can also be linked to amino termini of proteins is currently unclear. Here, the authors propose that cofilin-1 is sumoylated at its amino terminus and this regulates actin depolymerization. The concept is interesting, but the findings are largely based on overexpression experiments and could therefore represent artefacts that have no physiological relevance.

Comments

1. Virtually all experiments are carried out using overexpression of SUMO and overexpression of cofilin-1, which likely creates overexpression artefacts. The start of the results section is particularly weak because a potential overexpression artefact is not a sound base for the project. Mass spectrometry evidence for N-terminal sumoylation of endogenous cofilin-1 by endogenous SUMO is currently missing and is critical for the credibility of the manuscript.

2. As far as I am aware no previous project has ever found that endogenous cofilin-1 is sumoylated. The authors attempt to show endogenous sumoylation of cofilin-1 in Figure 1d, but the evidence is not convincing as the extra band in the co-immunoprecipitation could be a minor degradation product of the heavy chain, or a doublet of cofilin-1 because of a cysteine-bridge. Moreover, the cofilin-1 antibody appears not fully specific. Cropping of the blots is suboptimal as the relevant parts of the cofilin-1 blots are missing in the main figures. Does the reciprocal experiment work – IP of SUMO1,

blot for CFL1?

3. Modification of cofilin-1 and regulation of actin depolymerization are quite unexpected as SUMO is a nuclear protein. This raises the question of the physiological relevance of the findings. Is overexpressed SUMO in figure 4F also artificially located in the cytoplasm? The authors need to include overexpression of wild-type conjugatable and non-conjugatable SUMO with and without overexpressing cofilin-1 to test for potential overexpression artefacts of SUMO. Are endogenous cofilin-1 and actin depolymerization affected upon blocking endogenous SUMO signaling by E1 inhibiting compounds ML792 or TAK981?

4. Most amino termini of proteins are normally heavily acetylated. Is the N-terminus of endogenous cofilin-1 also acetylated? How is the acetyl group replaced by SUMO? The 'surprise' result that overexpressed cofilin-1 does not start at methionine, but at alanine is not trustworthy, since it could also represent an overexpression artefact.

5. Currently, it is unclear whether cofilin-1 is the only protein potentially regulated by N-terminal sumoylation. It would be interesting to use an unbiased approach to investigate whether other proteins are regulated in a similar manner – using endogenous SUMO and endogenous SUMO target proteins.

Here we provide a detailed point-by-point response to the reviewers' questions and explanation of how the new additions and revisions address their concerns, with relevant modifications indicated. We have printed the reviewers' comments in italics and our responses in plain text. All changes in the manuscript text are highlighted in blue.

Authors' Responses to the Reviewers' Comments:

NCOMMS-22-46870-T Decision Letter

Weng et al., **N-terminal α -amino SUMOylation of cofilin-1 is critical for its regulation of actin depolymerization**

We thank the reviewers for carefully evaluating our manuscript and providing us with an opportunity to respond to their suggestions and criticisms. As explained below in our point-by-point response, we have addressed each of the points raised through addition of supporting experiments, analyses, and text revision. Altogether, we think that these additional experiments and text revisions significantly improve the manuscript. Therefore, we would like to thank all the reviewers and editors for their time and input.

Reviewer #1(reviewers' comments in *italic*):

Summary of Critique: Cofilin-1, the ubiquitously expressed mammalian isoform, is here shown to be a substrate for SUMOylation, which would not in and of itself be surprising since cofilin has been characterized as a substrate for posttranslational modification (PTM) by ubiquitin and another ubiquitin-like modifier, NEDD-8. However, for these other cofilin PTMs, as well as all other known proteins that are substrates for SUMOylation, the modification occurs on epsilon-amino groups of lysine residues, whereas for cofilin the SUMO is on the N-terminal alpha-amino group. This reversible modification has implications for the phosphorylation of cofilin on its Ser3 residue, which is inhibitory to its interactions with both G- and F-actin. The authors go to great lengths using multiple methods to convincingly demonstrate this point. It is an unexpected finding because most cofilin in cells is N-acetylated on the penultimate alanine following its demethionylation, and, as noted herein, N-acetylation is an irreversible modification, suggesting that SUMOylation occurs on newly synthesized cofilin. Unfortunately, this aspect was not addressed through cell biological studies. What was shown is that SUMOylated cofilin binds somewhat better to F-actin than WT cofilin using in vitro co-sedimentation assays. The actual studies of SUMO-cofilin function in a cell biological context are the weakest part of this manuscript and rely on some "standard" methods that have serious interpretation problems when performed with cofilin-actin in complex

(addressed below).

We thank Reviewer #1 for her/his positive comments on our work, as well as the critical comments and insightful suggestions that have greatly helped us to improve our manuscript. As requested, we have performed multiple new experiments and reanalyzed our data to address the concerns raised.

Specific Points:

The intensity of the SUMOylated cofilin band on gels is very weak in Figure 1a,b and even in the prolonged exposure in Extended Figure 1. The authors do not address this until the discussion where they mention the small amount of SUMO-cofilin in explaining its oversized effect on actin dynamics suggesting its very rapid turnover, similar to what has been reported for NEDDylated cofilin on lysine 112. The authors should address this point up front when the blots are first presented so that the reader does not think small amounts of a modification might be insignificant in terms of functional consequences. It would be worthwhile to present the actual ratio of putative SUMO-cofilin to WT-cofilin in immunoblots of extracts from cells/tissues in Figure 3, but visualized using the same cofilin antibody, even though the band for SUMO-cofilin likely contains other mono-ubiquitylated, NEDDylated and SUMOylated species. Non-cofilin aficionados would probably benefit from seeing such a quantitative comparison to understand how small changes can have big impacts.

Response: We appreciate the reviewer's comment and have addressed the concern by providing an upfront explanation regarding the weak intensity of the SUMOylated CFL1 bands in the "Results" section, as suggested.

As mentioned in original manuscript, SUMOylation is known to be a highly dynamic post-translational modification, and the abundance of SUMOylated proteins is typically quite low (Becker J et al., 2013; Vertegaal ACO, 2022). Consequently, detecting SUMOylated CFL1 bands using CFL1 antibodies after immunoprecipitation of endogenous CFL1 poses a significant challenge. Even with overexpression of CFL1-HA and His-SUMO1 plasmids, an extended exposure time was necessary to visualize the SUMOylated CFL1 bands (Supplementary Figure 1). Despite multiple attempts, we were only able to identify the SUMOylated CFL1 bands in the immunoprecipitates pulled down by anti-CFL1 antibodies in a limited number of experiments. We have replaced the original results in Figure 3a-d with new results (Revised Figure 3a-d).

To address the request for a quantitative comparison, we have now included the actual ratio of putative SUMO-CFL1 to WT-CFL1 in the immunoblots of extracts from cells in Revised Figure 3a-d. However, it is important to note that the band corresponding to SUMOylated CFL1 likely contains other modified

products of CLF1 due to, for example, mono-ubiquitination, NEDDylation, and SUMOylation at other sites, if any. Nonetheless, as pointed out by the reviewer, providing this quantitative comparison will help readers, including non-experts in CFL biology and SUMO biology, to appreciate the large impact of these small changes on the protein function.

Reference:

Becker J, Barysch SV, Karaca S, Dittner C, Hsiao HH, Berriel Diaz M, Herzig S, Urlaub H, Melchior F. Detecting endogenous SUMO targets in mammalian cells and tissues. *Nat Struct Mol Biol.* 2013; 20(4): 525-31.

Vertegaal ACO. Signalling mechanisms and cellular functions of SUMO. *Nat Rev Mol Cell Biol.* 2022; 23(11):715-31.

Blots are presented with all of the appropriate controls but there is no quantification shown for any of the changes that seem to be significant. Since blots were repeated at least 3 times, there should be enough data available to provide graphs from which significance in the changes in SUMO-cofilin can be determined for Fig 1 a,b,c; Fig 2 d,f; and Fig 3. Also, in Figure 3b, why does control siRNA cause the seemingly large increase in SAE2 on the blot? This increase is certainly as significant as most other changes that are discussed.

Response: We appreciate the reviewer's comment and have made the necessary revisions to address the concerns.

To provide a more comprehensive analysis, we have now included quantifications along with the corresponding statistical values for Figure 1a-c, Figure 2d-f, and Figure 3, as well as the Revised Supplementary Figure 6. This addition will allow for a better assessment of the significance of the observed changes in SUMOylated CFL1 levels.

Regarding Figure 3b, we apologize for the oversight in selecting a representative Western blot image that may have led to confusion regarding the expression of SAE2 in the control siRNA sample. After re-evaluating the duplicate experiments and conducting an additional experiment, we found no evidence to suggest that control siRNA would cause an increase in SAE2 expression. Therefore, we have replaced the representative Western blotting image in Revised Figure 3b to accurately reflect the absence of any change in SAE2 expression with control siRNA. Thank you for bringing this to our attention, and we appreciate your suggestions in improving the clarity and accuracy of our findings.

With some quantifiable data included in the immunoblot figures, the biochemical and structural data presented in this manuscript all appear to be quite adequate and leave little room to doubt the conclusions that cofilin is SUMOylated on its N-terminus.

However, the functional significance of this finding with respect to how it affects in vitro actin dynamics and cell biological function needs to be better addressed.

Functional studies are all summarized in Figure 4. Co-sedimentation analyses with muscle F-actin strongly suggests that SUMO-cofilin binds better than WT to F-actin and thus appears in the pellet fraction in a higher ratio compared to WT cofilin. If these sedimentation studies were performed in the TLA-100 bench-top centrifuge in tubes holding 200 ul (as suggested but not specified in the methods), the authors should use the K value of the rotor and the speed to determine what size filaments might still be in solution after 30 minutes ($t=K/S$) where t is time, K is the rotor clearance value (usually obtained from the rotor booklet of the manufacturer) and S is the Svedberg units for the proteins in question. This was done for actin and ADF/cofilins in 2004 (doi: 10.1021/bi049797n) and it is surprising that even at 436,000xg (max speed used in this publication) it takes longer than 30 min to clear small pieces of cofilin-actin. At only 100,000xg as reportedly used here, larger fragments will still be in solution. This is an important point because of the cooperative binding of cofilin along a filament. Thus, regions of cofilin-saturated and “naked” F-actin occur together on a single filament, with severing often occurring at the junction between them. Is SUMO-cofilin still able to bind to F-actin in a cooperative manner or is there any steric effect of the SUMO on its binding to adjacent actins?

Response: We appreciate the reviewer's input and have taken note of the concerns raised regarding the co-sedimentation assay.

To address the cooperative binding and any potential steric effects of SUMO on the binding of adjacent actins, we performed the co-sedimentation assay using centrifugation conditions commonly employed in the literature. We referred to multiple studies to determine the appropriate centrifugation speed and duration. The references we consulted indicated a range of conditions, with most labs opting for a centrifugation speed of 100,000-150,000g for 20-30 minutes.

- 1) 100,000g×30min (Nakamura F et al., 1999)
- 2) 100,000g×30min (Srivastava J et al., 2008)
- 3) 100,000g×30min (Heier JA et al., 2017)
- 4) 150,000g×20min (Yoon J et al., 2017)
- 5) 150,000g×60min (Hung RJ et al., 2010)
- 6) 217,000g×20min (Kostan J et al., 2014)

In our study, we conducted the co-sedimentation assay under various

conditions, including centrifugation at 100,000g×30min, 100,000g×60min, 220,000g×30min, and 440,000g×30min. The results demonstrated that the different centrifugation conditions did not significantly affect the outcomes of the assay (Reviewer Figure 1). Based on these findings, we concluded that under the condition of 100,000g×30min, larger fragments of actin had already been centrifuged into the sediment.

While we did not explicitly calculate the size of the filaments remaining in solution after 30 minutes using the K value of the rotor and the Svedberg units, we followed established practices in the field and relied on the centrifugation conditions commonly employed for co-sedimentation assays involving actin and ADF/cofilins.

We acknowledge that further investigation into the cooperative binding of SUMO-CFL1 to F-actin and the potential steric effects of SUMO on its binding to adjacent actins would be valuable. These aspects could be explored in future studies to gain deeper insights into the functional consequences of CFL1 SUMOylation.

Thank you for highlighting these important considerations, and we appreciate your input in refining our understanding of the implications of our findings.

Reference:

Heier JA, Dickinson DJ, Kwiatkowski AV. Measuring Protein Binding to F-actin by Co-sedimentation. *J Vis Exp*. 2017; (123): 55613.

Hung RJ, Yazdani U, Yoon J, Wu H, Yang T, Gupta N, Huang Z, van Berkel WJ, Terman JR. Mical links semaphorins to F-actin disassembly. *Nature*. 2010; 463(7282): 823-7.

Kostan J, Salzer U, Orlova A, Törö I, Hodnik V, Senju Y, Zou J, Schreiner C, Steiner J, Meriläinen J, Nikki M, Virtanen I, Carugo O, Rappsilber J, Lappalainen P, Lehto VP, Anderluh G, Egelman EH, Djinić-Carugo K. Direct interaction of actin filaments with F-BAR protein pacsin2. *EMBO Rep*. 2014; 15(11): 1154-62.

Nakamura F, Huang L, Pestonjamas K, Luna EJ, Furthmayr H. Regulation of F-actin binding to platelet moesin in vitro by both phosphorylation of threonine 558 and polyphosphatidylinositides. *Mol Biol Cell*. 1999; 10(8): 2669-85.

Srivastava J, Barber D. Actin co-sedimentation assay; for the analysis of protein binding to F-actin. *J Vis Exp*. 2008; (13): 690.

Yoon J, Kim SB, Ahmed G, Shay JW, Terman JR. Amplification of F-Actin Disassembly and Cellular Repulsion by Growth Factor Signaling. *Dev Cell*. 2017; 42(2): 117-29.

One commonly used in vitro study that could go a long way toward answering some of these questions and is missing here is a study fluorescently tagged F-actin anchored on the surface of a coverslip while allowing various forms of cofilin to flow into the system. Both severing and end depolymerization could then be compared between SUMO- and

WT cofilin.

Response: We appreciate the reviewer's suggestion regarding the use of fluorescently tagged F-actin anchored on the surface of a coverslip to study the effect of different forms of cofilin on severing and end depolymerization. This approach could indeed provide valuable insights into the functional consequences of CFL1 SUMOylation.

During the revision period, we attempted to perform this experiment as suggested. We primarily referred to the following studies: Chen Q et al., 2011; Kang H et al., 2014; Chen Q et al., 2015; Chin SM et al., 2016; Shekhar S et al., 2017; Wioland H et al., 2019; Shekhar S et al., 2019; Wioland H et al., 2022; Henty-Ridilla JL, 2022.

However, we encountered several technical challenges as explained below that prevented us from completing it successfully:

- 1) Equipment limitations: This experiment requires a total internal reflection fluorescence (TIRF) microscope, which provides the needed signal-to-noise ratio due to the low penetration depth of the evanescent field. Unfortunately, neither our laboratory nor Shanghai Jiao Tong University had a TIRF microscope available for use. Instead, we attempted to use the Cell Observer from Zeiss to observe actin dynamics.
- 2) Lack of an appropriate flow chamber: We were unable to find commercial flow chambers suitable for our experiment. As a result, we tried to assemble one ourselves based on descriptions in the literature (Reviewer Figure 2). However, the initial flow chambers we constructed were not sealed properly which frequently leaked. This costed us a significant amount of time learning how to create a well-functioning flow chamber.
- 3) Challenges in anchoring fluorescently tagged F-actin: We made multiple attempts to anchor fluorescently tagged F-actin to the coverslip but were unsuccessful. We speculated that the issues might be related to inadequate coverage of the coverslip with Biotin-PEG or the inability of actin seeds to properly anchor to the coverslip. However, given the limited time available for the revision, we have not been able to find the solution and complete the suggested experiment.

We firmly believe that the findings from our current *in vitro* and *in vivo* experiments, combined with the existing literature, support the conclusion that N-terminal SUMOylation of CFL1 can regulate F-actin depolymerization. We are grateful for the reviewer's valuable suggestion and are committed to continue to explore conditions for performing the suggested experiment in our follow-up studies, which will definitely further enhance our understanding of the functional implications of CFL1 SUMOylation.

Reference:

Chen Q, Pollard TD. Actin filament severing by cofilin is more important for assembly than constriction of the cytokinetic contractile ring. *J Cell Biol.* 2011; 195(3): 485-98.

Chen Q, Courtemanche N, Pollard TD. Aip1 promotes actin filament severing by cofilin and regulates constriction of the cytokinetic contractile ring. *J Biol Chem.* 2015; 290(4): 2289-300.

Chin SM, Jansen S, Goode BL. TIRF microscopy analysis of human Cof1, Cof2, and ADF effects on actin filament severing and turnover. *J Mol Biol.* 2016; 428(8): 1604-16.

Henty-Ridilla JL. Visualizing Actin and Microtubule Coupling Dynamics In Vitro by Total Internal Reflection Fluorescence (TIRF) Microscopy. *J Vis Exp.* 2022; (185): 10.3791/64074.

Kang H, Bradley MJ, Cao W, Zhou K, Grintsevich EE, Michelot A, Sindelar CV, Hochstrasser M, De La Cruz EM. Site-specific cation release drives actin filament severing by vertebrate cofilin. *Proc Natl Acad Sci U S A.* 2014; 111(50): 17821-6.

Shekhar S, Carlier MF. Enhanced Depolymerization of Actin Filaments by ADF/Cofilin and Monomer Funneling by Capping Protein Cooperate to Accelerate Barbed-End Growth. *Curr Biol.* 2017; 27(13): 1990-1998.e5.

Shekhar S, Chung J, Kondev J, Gelles J, Goode BL. Synergy between Cyclase-associated protein and Cofilin accelerates actin filament depolymerization by two orders of magnitude. *Nat Commun.* 2019; 10(1): 5319.

Wioland H, Jégou A, Romet-Lemonne G. Celebrating 20 years of live single-actin-filament studies with five golden rules. *Proc Natl Acad Sci U S A.* 2022; 119(3): e2109506119.

Wioland H, Jegou A, Romet-Lemonne G. Quantitative Variations with pH of Actin Depolymerizing Factor/Cofilin's Multiple Actions on Actin Filaments. *Biochemistry.* 2019; 58(1): 40-7.

The studies on depolymerization of pyrene-F-actin also require careful interpretation. Cofilin binding to pyrene-F-actin causes quenching of the pyrene fluorescence which can be misconstrued as depolymerization. Thus, a second method to confirm the depolymerization is required. The enhanced binding of SUMO-cofilin to F-actin shown in Fig 4 C might account for its apparent “stimulation of depolymerization” in Fig 4D, but may actually arise from quenching of fluorescence, although if SUMO-cofilin still binds cooperatively, the time course of 3 min for complete loss of pyrene signal might not fit with the rate of binding. The authors need to examine the rates for both binding and fluorescence quenching from previous publications on WT cofilin.

Response: We appreciate the reviewer's valuable input regarding the interpretation of the depolymerization studies using pyrene-F-actin. We agree that the quenching of pyrene fluorescence upon CFL1 binding can be misconstrued as depolymerization, and it would be important to employ

additional methods to confirm the depolymerization process.

As suggested, we have compared the rate of cofilin-induced F-actin depolymerization measured by FRET as shown by Nadkarni AV and Briehner WM (2014) with our result. We traced the effect of 2.5 μM cofilin (closest to our use of 5 μM CFL1) on fluorescence transfer during the time period of 0 to 600 s (Figure 5A, green circles of Nadkarni AV and Briehner WM, 2014) and overlaid the curve on our Figure 4e). We found that kinetic wise, their time course matches nearly perfectly with what we determined for CFL1^{WT} and CFL2^{KR} (Reviewer Figure 3). By contrast, the kinetics for cofilin quenching of pyrenyl actin fluorescence was much faster, reaching maximum in < 70 s (Figure S2C of Nadkarni AV and Briehner WM, 2014, traced curve overlaid in Reviewer Figure 3 for comparison). In an early paper by Carlier MF et al. (1997) the quenching of pyrenyl actin by ADF was also shown to reach the maximum in < 70 s (Figure 3a, b of the Carlier paper). In addition, dynamic observations of F-actin using TIRF microscopy have shown that F-actin severing by CFL1 occurs within seconds to tens of seconds (Kang H et al., 2014; Wioland H et al., 2019). Therefore, although quenching can contribute to fluorescence decrease of pyrenyl actin, the kinetics of fluorescence changes we observed are more in line with the rate of cofilin-induced F-actin depolymerization in the literature monitored using either pyrenyl actin (for example, Figure 3B in Moriyama K and Yahara I, 2002) or other methods independently of pyrenyl actin.

We appreciate the reviewer's insights and we believe that the combination of our experimental findings and the literature references on CFL1 severing and depolymerization rates collectively support our conclusions regarding F-actin depolymerization.

Reference:

Carlier MF, Laurent V, Santolini J, Melki R, Didry D, Xia GX, Hong Y, Chua NH, Pantaloni D. Actin depolymerizing factor (ADF/cofilin) enhances the rate of filament turnover: implication in actin-based motility. *J Cell Biol.* 1997; 136(6): 1307-22.

Kang H, Bradley MJ, Cao W, Zhou K, Grintsevich EE, Michelot A, Sindelar CV, Hochstrasser M, De La Cruz EM. Site-specific cation release drives actin filament severing by vertebrate cofilin. *Proc Natl Acad Sci U S A.* 2014; 111(50): 17821-6.

Moriyama K, Yahara I. Human CAP1 is a key factor in the recycling of cofilin and actin for rapid actin turnover. *J Cell Sci.* 2002;115(Pt 8):1591-601.

Nadkarni AV, Briehner WM. Aip1 destabilizes cofilin-saturated actin filaments by severing and accelerating monomer dissociation from ends. *Curr Biol.* 2014; 24(23): 2749-57.

Wioland H, Jegou A, Romet-Lemonne G. Torsional stress generated by ADF/cofilin on cross-linked actin filaments boosts their severing. *Proc Natl Acad Sci U S A.* 2019; 116(7): 2595-602.

Finally, the data in Figure 4F assumes that fluorescent phalloidin binds to all cellular filamentous actin and thus its decreased degree of labeling

shows loss of F-actin. However, phalloidin does not bind to cofilin-actin. Furthermore, using Triton as the cell permeabilization agent prevents cofilin immunolabeling of cofilin-actin filaments. Together, these events have led to a decade of misinterpretation of decreased phalloidin binding as meaning less assembled actin filaments. This information concerning cofilin-actin been in the literature for many years but is widely ignored since Triton permeabilization is the “go to” method in virtually every immunolabeling textbook. To visualize cofilin-actin filaments by cofilin immunolabeling requires permeabilization using either 100% -20o C methanol or acetone. Acetone permeabilization allows some phalloidin staining but even so, phalloidin does not stain cofilin-actin filaments. A recent paper demonstrating this very well in the proximal filopodia of neuronal growth cones was published by Hylton et al. (<https://doi.org/10.1038/s41467-022-30116-x>) with a follow up methods paper (DOI: 10.1007/978-1-0716-2811-9_18). Thus, the loss of phalloidin staining could arise through enhanced SUMOcofilin-stabilized F-actin which competes with myosin II for actin binding and thus reduces tension in cells through actin cables; competition between cofilin and myoII binding regulates cortical and nuclear tension in normal cell behavior. It could also be due to very specific effects on focal adhesions- suggested by the cell rounding. These adhesions may be sites of local cofilin synthesis which could be the major targets for SUMOylation of newly synthesized cofilin and worth looking at in detail. Given this information, the data in Fig 4F is really not very useful without additional panels demonstrating that there is not and increase of cofilin-actin filaments. Staining focal adhesions in spread cells and observing their change during SUMOylation of cofilin would also be a worthwhile addition. Cell rounding might result from a direct effect of SUMO on actin depolymerization and not adhesion, but this is hard to address without being able to acutely turn on SUMO, something that could be done in future studies. Avoid studying cofilin-actin filaments in vivo using Lifeact or the other F-actin binding peptide chimeras with fluorescent proteins- these do not visualize cofilin-actin filaments because they recognize the same filament conformation as phalloidin.

Response: We appreciate the valuable suggestions and insights provided by the reviewer. The information shared regarding the binding of phalloidin to cofilin-actin filaments and the challenges associated with Triton X-100 permeabilization is indeed important. The suggestions on SUMOylated cofilin stabilizing F-actin and competing with myosin II to reduce tension leading to cell rounding, the potential effect of SUMOylated cofilin in focal adhesion and other possible interpretations of our data are all very thoughtful and definitely helpful to our design of follow-up studies to further evaluate the role of N- α -SUMOylation on CFL1.

Hylton et al. analyzed the effects of different fixing reagents and different permeabilizing reagents on phalloidin staining and cofilin staining. They found that PFA fixation + Triton X-100 permeabilization disrupts cofilin-actin bundle staining but does not affect phalloidin staining of F-actin; PFA fixation + acetone permeabilization slightly inhibits the phalloidin signal; while methanol fixation disrupts phalloidin binding to F-actin (Hylton RK et al., 2022). Therefore, there appears to be no absolute certainty that any of the fixation methods is suitable for detecting cofilin-actin. Since we found that the N-terminal SUMOylation of CFL1 regulates F-actin depolymerization in the *in vitro* F-actin depolymerization assay, we focused more on the level of F-actin in cells after changing the SUMOylation level of CFL1. For this reason, we chose PFA fixation + Triton X-100 permeabilization. Collectively, the F-actin co-sedimentation assay, actin depolymerization assay and F-actin immunofluorescence staining in Figure 4 together support our conclusion that the N-terminal SUMOylation of CFL1 is involved in regulating F-actin depolymerization. No single experiment could independently draw such a conclusion.

We appreciate the reviewer's input, and we will take these points into consideration in future investigations. The information provided helps us approach our findings with greater caution and promotes a more comprehensive understanding of the relationship between cofilin-actin filaments and SUMOylation of CFL1. In the revised discussion, we have added a new paragraph stating that caution should be taken when interpreting our data, taking account of the cofilin effects on pyrene fluorescence and phalloidin staining.

Finally, we would like to emphasize that all research projects have different phases of development. As the reviewer correctly pointed earlier, the current study focuses mainly on demonstrating CFL1 SUMOylation and identifying the SUMOylation site. We made an important and novel finding that this PTM can occur at the N-terminus of a protein. Our data in Figure 4b,c clearly show that CFL1 SUMOylation affects its ability to bind to F-actin, and that in Figure 4d-f are consistent with the idea that this binding is able to decrease F-actin levels. The most straight forward interpretation is that SUMOylation facilitates F-actin depolymerization, although there are other possible interpretations as suggested by the reviewer. At the current stage, however, these interpretations are rather speculative and will need a lot of experimental work to evaluate, especially given the need to develop and optimize the methodology in order to properly address some of the suggested possibilities. Therefore, it would be most appropriate to evaluate them in future studies that focus on various ways by which SUMOylation can affect CFL1 regulation of actin dynamics and/or any actin-independent functions of CFL1.

Reference:

Hylton RK, Heebner JE, Grillo MA, Swulius MT. Cofilin filaments regulate filopodial structure and dynamics in neuronal growth cones. *Nat Commun.* 2022; 13(1): 2439.

The authors are certainly aware that cofilin disassembly of F-actin in vivo involves the interplay of many other proteins, some of which could be strongly affected by SUMO-cofilin. Among these are AIP1 (aka WDR1), which enhances severing of cofilin-actin, and essential cyclase-associated proteins (CAP1/2) which binds cofilin-actin filaments releasing cofilin and enhancing nucleotide exchange on the ADP-actin (to ATP-actin) as it is released. While studying how SUMO-cofilin affects the function of AIP1 or CAPs is not an expectation for this manuscript, acknowledging the broader aspects of SUMO-cofilin involving other proteins would be a useful addition to the discussion.

Response: We appreciate the reviewer's insightful comment regarding the interplay between CFL1 and other proteins involved in F-actin disassembly in vivo. You are correct in pointing out that proteins such as AIP1 (WDR1) and cyclase-associated proteins (CAP1/2) play important roles in regulating CFL1-actin interactions and actin dynamics. While our study did not specifically investigate the impact of SUMOylated CFL1 on the function of AIP1 or CAPs, we acknowledge the broader context of SUMOylated CFL1 involvement in relation to these proteins.

In the revised manuscript, we now include a section in the Discussion that highlights the potential interactions and crosstalks between SUMOylated CFL1 and other actin-binding proteins involved in F-actin disassembly. We emphasize that future studies should explore the specific impact through which SUMOylated CFL1 may modulate the function of AIP1, CAPs, and other relevant proteins to gain a more comprehensive understanding of the regulation of CFL1-mediated F-actin dynamics.

We thank the reviewer for bringing this important aspect to our attention, and we believe that including this information in the discussion will enhance the overall interpretation and context of our findings.

Reviewer #2:

In this manuscript, the authors report on their discovery of N-terminal conjugation of a substrate protein (CFL1) with the SUMO1 peptide. First, they show that CFL1 is modified by SUMO1 ex vivo in cultured cells, in vitro in a reconstituted sumoylation system, and also in vivo in mouse brains. Sumoylation ex vivo seems to be reversible upon overexpression of the SUMO protease SENP1, but not of a catalytically inert SENP1 mutant. Then, using multiple complementary approaches, they demonstrate that this modification occurs on the N-terminal amino group of CFL1, rather than on any of the internal lysines, which is indeed a novel

and interesting finding.

To identify the major SUMO1 conjugation site on CFL1, the authors mutate each of the 25 lysine residues on CFL1 into arginines, and also create a 25KR mutant where all 25 lysines are mutated. Even though the K112R and K114R mutants exhibit loss of sumoylation to a certain degree, none of the mutants, including the 25KR mutant display total loss of sumoylation, suggesting that CFL1 could be modified outside one of these lysines..

They perform mass spectrometry analyses on endogenous CFL1 (pulled-down from Neuro2A cells) by taking advantage of an overexpressed His-SUMO1E93R mutant. MS/MS data revealed that the major conjugation site carrying the QTTG signature following tryptic cleavage was the N-terminal alpha amino group. Protection of this group by acetylation (upon Naa60 overexpression) abolished CFL1 sumoylation. They also perform methylation assays, to “methyl-protect” or “methyl-block” all of the internal lysine epsilon amino groups, or those plus the N-terminal alpha amino group, then perform in vitro sumoylation. These experiments also validated that the free internal lysine side chains were dispensible for SUMO1 conjugation, but not the amino end.

Finally, the authors perform several functional experiments to demonstrate that sumoylated CFL1 (either in vitro sumoylated CFL1, or an engineered version of CFL1 with an N-terminal SUMO1 tag that mimics its conjugation) binds more efficiently to F-actin compared to non-sumoylated CFL1 and induces depolymerization more efficiently.

Overall, these are novel and interesting findings. Firstly, a growing number of proteins now appear to be sumoylated outside of the traditional sumoylation consensus motif ψ -K-X-D/E, and these non-traditional conjugation sites are sometimes difficult to identify. In such cases where finding the major SUMO conjugation site proves to be difficult, this manuscript now suggests that researchers in the field should also consider the possibility that a substrate may be modified at the N-terminus. Conjugation of ubiquitin to alpha amino groups has been reported, but this was previously unrecognized for the SUMO field. Secondly, although CFL1 is known to be regulated by multiple PTMs, this manuscript now unveils another layer of regulation for this important substrate.

We thank Reviewer #2 for acknowledging the novelty and importance of our work, constructive comments and encouraging assessment.

In conclusion, I recommend this well-written and nicely organized manuscript be published provided that the authors also address the following comments:

1) It is rather confusing that the authors use different cell lines throughout the manuscript. For example, some of the cell-based experiments in Fig 1 were performed in CHO-K1 cells. Then, the mass spec results on endogenous CFL1 were obtained using the Neuro2A cell line. In Figure 3, however, SUMO E1, E2 enzyme knock-downs or the experiments probing physical interactions between CFL1 and E1 subunits (or E2) were all performed in HEK293 cells. Why is that? How do the authors justify this non-uniformity? It is certainly OK and even advisable to repeat an experiment and validate its result in a second cell system; however, employing different cell lines for different experiments should be avoided, or at least justified. Also, is there a particular reason why the authors use mouse cerebral cortexes for the experiment in Fig 1D?

Response: We thank the reviewer for pointing out the inconsistency in cell lines used throughout the manuscript and apologize for any confusion this had caused. We used different cell lines for various experiments based on their origin, growth characteristics, transfection efficiency, protein expression, and unspecific applications. Here is the justification for employing different cell lines:

We have demonstrated that endogenous CFL1 can be SUMOylated in Neuro-2a and HEK-293T cells (Figure 2b, Figure 3a-d). In Reviewer Figure 4, we also show that endogenous CFL1 can be modified by SUMO1 in CHO-K1 cells, demonstrating the conservation of CFL1 SUMOylation in different cell lines across different species.

Since we used mouse brain to demonstrate endogenous SUMOylation of CFL1 in vivo, we thought it to be also important to determine the SUMO-conjugated site(s) using mouse endogenous CFL1, to avoid potential issues arising from species differences. Therefore, we used mouse-derived Neuro-2a cells, as shown in Figure 2a-b. These neuroblastoma cells also have some characteristics of neurons, making them more appropriate as a surrogate for mouse brain.

The use of HEK-293T cells was due to the availability of the Ubc9 antibodies in our lab, which recognizes human Ubc9 better. Therefore, to investigate the protein interactions between CFL1 and endogenous SAE1, SAE2, Ubc9, and SENP1, this human cell line was utilized, as presented in Figure 3a-d.

CHO-K1 cells are commonly employed for transfection experiments due to their high transfection efficiency. We used CHO-K1 cells to perform biological experiments that required cotransfection different CFL1 mutations with other proteins. Additionally, CHO-K1 cells adhere well to coverslips, allowing us to study actin morphology more easily in immunofluorescence experiments.

Regarding why we use mouse cerebral cortices for the experiment in Figure 1d, this is due to the fact that CFL1 is expressed in various tissues and cells, with higher expression levels observed in the brain and liver (Vartiainen MK et al., 2002).

In summary, we selected different cell lines based on their suitability for specific experimental objectives, based on factors such as species specificity of the antibodies, transfection efficiency, ease of protein detection, and tissue relevance. We acknowledge the importance of providing clear justifications for the choice of cell lines and made additional revisions to convey such information in the revised manuscript.

Reference:

Vartiainen MK, Mustonen T, Mattila PK, Ojala PJ, Thesleff I, Partanen J, Lappalainen P. The three mouse actin-depolymerizing factor/cofilins evolved to fulfill cell-type-specific requirements for actin dynamics. *Mol Biol Cell*. 2002;13(1):183-94.

2) It would be interesting to determine if CFL1 also undergoes SUMO2/3 conjugation, and whether that also occurs at the N-terminal alpha amino group. It should be relatively easy to repeat some of the simple experiments (i.e. Fig 1A, 2C, 2D, 2G) using His-SUMO2/3.

Response: We appreciate the reviewer's point on this important issue. In response to the suggestion, we have investigated SUMO2/3 conjugation of CFL1. Since SUMO2 and SUMO3 share a high level of similarity (97% identical), we primarily focused on studying SUMO2 conjugation of CFL1.

As shown in Reviewer Figure 5, CFL1 was found to undergo SUMO2 modification. Similar to CFL1 SUMO1 conjugation, only one SUMOylated band was observed when CFL1 was co-expressed with His-SUMO2, as detected by anti-SUMO2/3 antibody (Reviewer Figure 5a). Moreover, even when all lysine residues in CFL1 were mutated to arginines (25KR), CFL1 still exhibited SUMO2 modification (Reviewer Figure 5b). Additionally, when CFL1 was co-expressed with Naa60, the SUMO2 conjugation of CFL1 was inhibited (Reviewer Figure 5c). Furthermore, after *in vitro* SUMO2 conjugation, the SUMOylated bands were not detectable in the N-terminal GST-tagged CFL1 protein (Reviewer Figure 5d). These results support the possibility of N-terminal SUMO2/3 conjugation of CFL1. However, further confirmation of the N-terminal SUMO2/3 conjugation of CFL1 would require mass spectrometry analysis in future study.

3) In all IP (immunoprecipitation) experiments, starting from Fig 1A, the band corresponding to sumoylated CFL1 is extremely weak and barely visible. Indeed, for most protein substrates, only a small portion of the entire pool gets modified and the authors discuss this issue. However, this seems to be extreme for CFL1. Because the authors are using His-tagged SUMO1 in some cases (i.e. Fig 1A), it would be nice to perform an

“anti-His” Western blot on their immunoprecipitated samples. One would wonder if that yields a better signal.

Response: We appreciate the reviewer's suggestion. As requested, we conducted additional experiments to address this issue. CHO-K1 cells were transiently transfected with CFL1-HA plasmid, with or without His-SUMO1. The cell lysates were subjected to immunoprecipitation using an anti-HA antibody, followed by Western blotting with an anti-His antibody (1:1000, 66005-1-Ig, Proteintech).

As depicted in Reviewer Figure 6, the intensity of the SUMO1-modified CFL1 bands detected by the anti-His antibodies was actually weak. Our explanation is that SUMOylation is considered a highly dynamic post-translational modification, and the proportion of SUMOylated proteins at any given time is extremely low (Becker J et al., 2013; Vertegaal ACO, 2022). Therefore, it is quite difficult to identify SUMOylated CFL1 band by anti-SUMO antibodies or anti-tag antibodies after immunoprecipitating CFL1.

Reference:

Becker J, Barysch SV, Karaca S, Dittner C, Hsiao HH, Berriel Diaz M, Herzig S, Urlaub H, Melchior F. Detecting endogenous SUMO targets in mammalian cells and tissues. *Nat Struct Mol Biol.* 2013; 20(4): 525-31.

Vertegaal ACO. Signalling mechanisms and cellular functions of SUMO. *Nat Rev Mol Cell Biol.* 2022; 23(11):715-31.

4) In Fig 3A, the IP seems to have been performed on endogenous proteins (endogenous CFL1 and endogenous SUMO1). This experiment, compared to the experiment in Fig 1A where SUMO1 is overexpressed, yields a sumoylated CFL1 band that is of the same weak intensity. Shouldn't the authors detect enhanced sumoylation of CFL1 when they overexpress SUMO1, as in Fig 1A? Does that happen only in the presence of UBC9?

Response: We appreciate the reviewer's suggestion and insightful points. The intensities of bands in Western blotting images are influenced by various factors, including exposure duration and the number of times the primary antibodies are used. Therefore, it is best to compare band intensities in Western blotting images on the same transfer membrane with the same exposure. It is not possible to compare band intensities between different Western blotting images.

In Supplementary Figure 3b, endogenous SUMO1 was responsible for the SUMOylation of overexpressed CFL1. Due to the low level of SUMOylation, the SUMOylated band in the Western blotting was not clearly visible. This situation changed when His-SUMO1 was added, allowing the SUMOylated band to be readily detectable, and upon coexpression of Myc-Ubc9, the band intensity was further enhanced. These results are totally consistent with the notion that

SUMOylation is very dynamic and the balance can be tipped towards more SUMOylation by increasing SUMO conjugation.

5) Figure 3D: in the text, the authors say “knockdown of SENP1 markedly increases the SUMOylation levels of CFL1”. The results shown in this Western blot are not strong enough to support this conclusion. I can barely see a band, let alone a remarkable increase in the band’s intensity. How many times has this experiment been performed? Is this the most representative WB? Data here do not support the conclusion.

Response: We thank the reviewer for this observation. For Figure 3d, we conducted three independent experiments. In response to the concern raised, we repeated the experiment. As depicted in Revised Figure 3d, the level of CFL1 SUMOylation increased approximately twofold after the knockdown of SENP1. Although direct evidence for the compensation of other SENPs in the absence of SENP1 is lacking, several studies have reported that SENP1 and other SENPs can deSUMOylate the same substrates, such as amyloid precursor protein, androgen receptor, and pin1 (Kaikkonen S et al., 2009; Chen CH et al., 2013; Maruyama T et al., 2018). Therefore, we speculate that other SENPs might partially compensate for the loss of SENP1, and the increase in CFL1 SUMOylation levels may not be particularly pronounced.

Additionally, following the suggestion of Reviewer #1, we have replaced Figure 3d with the Revised Figure 3d.

Reference:

Chen CH, Chang CC, Lee TH, Luo M, Huang P, Liao PH, Wei S, Li FA, Chen RH, Zhou XZ, Shih HM, Lu KP. SENP1 deSUMOylates and regulates Pin1 protein activity and cellular function. *Cancer Res.* 2013; 73(13): 3951-62.

Kaikkonen S, Jääskeläinen T, Karvonen U, Rytinki MM, Makkonen H, Gioeli D, Paschal BM, Palvimo JJ. SUMO-specific protease 1 (SENP1) reverses the hormone-augmented SUMOylation of androgen receptor and modulates gene responses in prostate cancer cells. *Mol Endocrinol.* 2009; 23(3): 292-307.

Maruyama T, Abe Y, Niikura T. SENP1 and SENP2 regulate SUMOylation of amyloid precursor protein. *Heliyon.* 2018; 4(4): e00601.

6) SENP1 is a predominantly nuclear protease. How do the authors explain its “desumoylase” effect on CFL1, a cytoplasmic protein?

Response: We thank the reviewer for raising this important and intriguing point. The predominantly nuclear localization of SENP1 is well established. However, it is worth noting that SENP1 has the ability to localize both in the nucleus and the cytoplasm. This dual localization is facilitated by a specific motif called the nuclear export sequence (NES) located at the C-terminus of the SENP1 protein (Kim YH et al., 2005; Bailey D et al., 2004). The presence of this NES motif allows for the export of SENP1 from the nucleus to the cytoplasm.

It is important to mention that the presence of the NES motif is not unique to SENP1. Other members of the SENP family, such as SENP2 and SENP3, also possess this motif (Kim YH et al., 2005; Guo C et al., 2014). The conservation of this motif between SENP1 in mice and humans suggests its potentially crucial role in the function of these proteins.

Given the dual localization capability of SENP1, it is possible that the "desumoylase" effect on CFL1, a cytoplasmic protein, is mediated by SENP1's presence in the cytoplasm. However, further research is necessary to fully elucidate the precise mechanisms and regulation of this process.

To provide additional support for the ability of SENPs to deSUMOylate extra-nuclear proteins, we have compiled a table listing examples of extra-nuclear proteins that have been shown to undergo SUMOylation and deSUMOylation by SENPs. These studies further support the notion that SENPs can target and regulate the SUMOylation status of proteins outside the nucleus.

Proteins	Locations	Reference
PTP1B	cytoplasm	Dadke S et al., Nat Cell Biol , 2007
Synapsin Ia	cytoplasm	Tang LT et al., Nat Commun , 2015
ANXA1	cytoplasm	Li X et al., Sci Adv , 2021
PDPK1	cytoplasm	Hu B et al., Autophagy , 2021
PKC α	cytoplasm	Sun H et al., Nat Commun , 2014
PKC δ	cytoplasm	Gao S et al., FEBS J , 2021
PKC ϵ	cytoplasm	Zhao X et al., Cell Rep , 2020
NEMO/IKK γ	cytoplasm	Desterro JM et al., Mol Cell , 1998
AMPK	cytoplasm	Rubio T et al., Mol Biol Cell , 2013
RIM1 α	cytomembrane	Girach F et al., Cell Rep , 2013
TRPV1	cytomembrane	Wang Y et al., Nat Commun , 2018
K ₂ P ₁	cytomembrane	Rajan S et al., Cell , 2005
K _{v1.5}	cytomembrane	Benson MD et al., Proc Natl Acad Sci U S A , 2007
K _{v7.2}	cytomembrane	Qi Y et al., Neuron , 2014

Reference:

Bailey D, O'Hare P. Characterization of the localization and proteolytic activity of the SUMO-specific protease, SENP1. *J Biol Chem*. 2004; 279(1): 692-703.

Benson MD, Li QJ, Kieckhafer K, Dudek D, Whorton MR, Sunahara RK, Iñiguez-Lluhí JA, Martens JR. SUMO modification regulates inactivation of the voltage-gated potassium channel Kv1.5. *Proc Natl Acad Sci U S A*. 2007; 104(6): 1805-10.

Dadke S, Cotteret S, Yip SC, Jaffer ZM, Haj F, Ivanov A, Rauscher F 3rd, Shuai K, Ng T, Neel BG, Chernoff J. Regulation of protein tyrosine phosphatase 1B by sumoylation. *Nat Cell Biol*. 2007; 9(1): 80-5.

Desterro JM, Rodriguez MS, Hay RT. SUMO-1 modification of I κ B α inhibits NF- κ B activation. *Mol Cell*. 1998; 2(2): 233-9.

Gao S, Zhao X, Hou L, Ma R, Zhou J, Zhu MX, Pan SJ, Li Y. The interplay between SUMOylation and

phosphorylation of PKC δ facilitates oxidative stress-induced apoptosis. *FEBS J.* 2021; 288(22): 6447-64.

Girach F, Craig TJ, Rocca DL, Henley JM. RIM1 α SUMOylation is required for fast synaptic vesicle exocytosis. *Cell Rep.* 2013; 5(5): 1294-301.

Guo C, Henley JM. Wrestling with stress: roles of protein SUMOylation and deSUMOylation in cell stress response. *IUBMB Life.* 2014; 66(2): 71-7.

Hu B, Zhang Y, Deng T, Gu J, Liu J, Yang H, Xu Y, Yan Y, Yang F, Zhang H, Jin Y, Zhou J. PDPK1 regulates autophagosome biogenesis by binding to PIK3C3. *Autophagy.* 2021; 17(9): 2166-83.

Kim YH, Sung KS, Lee SJ, Kim YO, Choi CY, Kim Y. Desumoylation of homeodomain-interacting protein kinase 2 (HIPK2) through the cytoplasmic-nuclear shuttling of the SUMO-specific protease SENP1. *FEBS Lett.* 2005; 579(27): 6272-8.

Li X, Xia Q, Mao M, Zhou H, Zheng L, Wang Y, Zeng Z, Yan L, Zhao Y, Shi J. Annexin-A1 SUMOylation regulates microglial polarization after cerebral ischemia by modulating IKK α stability via selective autophagy. *Sci Adv.* 2021; 7(4): eabc5539.

Qi Y, Wang J, Bomben VC, Li DP, Chen SR, Sun H, Xi Y, Reed JG, Cheng J, Pan HL, Noebels JL, Yeh ET. Hyper-SUMOylation of the Kv7 potassium channel diminishes the M-current leading to seizures and sudden death. *Neuron.* 2014; 83(5): 1159-71.

Rajan S, Plant LD, Rabin ML, Butler MH, Goldstein SA. Sumoylation silences the plasma membrane leak K⁺ channel K2P1. *Cell.* 2005; 121(1): 37-47.

Rubio T, Vernia S, Sanz P. Sumoylation of AMPK β 2 subunit enhances AMP-activated protein kinase activity. *Mol Biol Cell.* 2013; 24(11): 1801-11, S1-4.

Sun H, Lu L, Zuo Y, Wang Y, Jiao Y, Zeng WZ, Huang C, Zhu MX, Zamponi GW, Zhou T, Xu TL, Cheng J, Li Y. Kainate receptor activation induces glycine receptor endocytosis through PKC deSUMOylation. *Nat Commun.* 2014; 5:4980.

Tang LT, Craig TJ, Henley JM. SUMOylation of synapsin Ia maintains synaptic vesicle availability and is reduced in an autism mutation. *Nat Commun.* 2015; 6:7728.

Wang Y, Gao Y, Tian Q, Deng Q, Wang Y, Zhou T, Liu Q, Mei K, Wang Y, Liu H, Ma R, Ding Y, Rong W, Cheng J, Yao J, Xu TL, Zhu MX, Li Y. TRPV1 SUMOylation regulates nociceptive signaling in models of inflammatory pain. *Nat Commun.* 2018; 9(1): 1529.

Zhao X, Xia B, Cheng J, Zhu MX, Li Y. PKC ϵ SUMOylation Is Required for Mediating the Nociceptive Signaling of Inflammatory Pain. *Cell Rep.* 2020; 33(1): 108191.

7) The physical association between CFL1 and the E1 enzyme subunits is interesting and confusing at the same time, and deserves further exploration and validation. The authors should make an effort to prove this potential physical interaction using a complementary approach, for example, proximity ligation assays (PLA). If the authors end up employing PLA, they should also try to detect CFL1 / SENP1 interactions.

Response: We appreciate the reviewer's excellent suggestion and insightful points. In response to the reviewer's request, we have conducted additional experiments to further validate the potential physical interactions between CFL1 and the E1 enzyme subunits using the proximity ligation assay (PLA).

As shown in Reviewer Figure 7, we performed PLA using antibodies against endogenous CFL1, SAE1, SAE2, and SENP1 in CHO-K1 cells. The red signal, indicative of protein-protein interaction, was detected when CHO-K1 cells were incubated with both SAE1 and CFL1 antibodies, supporting the interaction between CFL1 and SAE1. Similarly, positive PLA signals were observed for CFL1 and SAE2, as well as CFL1 and SENP1, indicating their physical interactions.

The PLA experiment was conducted more recently because the PLA kit had to be imported from the United States. There was an unexpected delay in transportation and customs clearance for this kit, although we promptly placed the order upon receiving the editor's email. For this reason, we have only had a limited number of repeats and have not tested all interacting partners, e.g., Ubc9 with CFL1.

We thank the reviewer for suggesting this valuable experiment, and we believe that the PLA results strengthen the evidence for the physical interactions of CFL1 with the E1 enzyme subunits and SENP1.

8) In Figure 4F, the authors use CHO-K1 cells where the endogenous CFL1 is stably knocked down. The labeling on the first panel (far left in the figure) "without CFL1 KD" is confusing. They should just write "cells without CFL1" or "CFL1 KD cells".

Response: We thank the reviewer for pointing out the confusion in the labeling of the first panel in Figure 4f. We have carefully considered your suggestion, and modified the labeling accordingly. Instead of "WT CHO-K1 cells without CFL1 KD", we revised it to simply "WT CHO-K1 cells". This modification will help clarify the labeling and eliminate any confusion.

9) Again, in Figure 4F, it is difficult, if not impossible, to see a difference in the phalloidin signals between i (or ii) and iii, contrary to what the authors claim. Is it possible to quantify these signals? The authors should use a software to quantify signals from at least 20-30 cells for each experimental group, and represent the results on a graph (along with the averages and error bars). Without quantifications, it is hard to make comparisons and draw conclusions. For example, why is it that CFL25KR (viii), which is sumoylated as well as CFL-WT (iii), seems to be less effective? Why does it need UBC9 and SUMO1 co-expression (ix) to be only as effective as CFL-WT without UBC9/SUMO1?

Response: We appreciate the reviewer's valuable suggestion. As requested, we have incorporated the quantitative analysis in the Revised Supplementary Figure 6.

Regarding the comparison between CFL1^{25KR} (viii) and CFL1^{WT} (iii), since CFL1^{25KR} contains the K112R and K114 mutations and the SUMOylation level of CFL1^{2KR} (K112R/K114R) was lower than that of CFL1^{WT} (Supplementary Figure 3a), it is expected that CFL1^{25KR} (viii) SUMOylation level is lower than that of CFL1^{WT} (iii). We have also demonstrated the aforementioned outcome in the same Western blotting image (Reviewer Figure 8). This may explain why the effect of CFL1^{25KR} on the phalloidin signal appears weaker compared to CFL1^{WT}.

In the case of CFL1^{25KR}+SUMO1+Ubc9 (ix), the SUMOylation level of CFL1^{25KR} significantly increased when it was co-expressed with Ubc9 and SUMO1. This enhanced SUMOylation promoted F-actin depolymerization, leading to the decrease of the phalloidin signal. In the original manuscript, the comparison between CFL1^{25KR} (viii) and CFL1^{25KR} +SUMO1+Ubc9 (ix) partially supports the role of CFL1 N-terminal SUMOylation in F-actin depolymerization.

However, we caution against directly comparing CFL1^{25KR} +SUMO1+Ubc9 (ix) with CFL1^{WT} (iii) for two reasons: First, the CFL1^{25KR} mutant contains multiple lysine residue changes, which could significantly alter the protein structure and function of CFL1. Second, lysine residues are sites for various post-translational modifications (PTMs) on proteins, and extensive lysine mutations may disrupt CFL1 function by affecting PTMs. Therefore, CFL1^{25KR} +SUMO1+Ubc9 (ix) is only suitable for comparison with CFL1^{25KR} (viii). We cannot solely attribute the observed differences between CFL1^{WT} (iii) and CFL1^{25KR} +SUMO1+Ubc9 (ix) to N-terminal SUMOylation of CFL1.

We have considered these points and their implications when interpreting our results in the manuscript. Thank you for raising these important concerns, and we appreciate your insightful feedback.

10) Again, in Figure 4F, the authors can add a panel where they treat cells with a sumoylation inhibitor (i.e. ML792).

Response: We appreciate the reviewer's valuable suggestion. To address this concern, we conducted additional experiments as requested. Specifically, we treated both WT CHO-K1 cells and endogenous CFL1-knockdown cells expressing CFL1^{WT}-HA with 10 μ M of ML-792 for 24 hours. As shown in Reviewer Figure 9, the F-actin levels were significantly reduced after ML-792 treatment, which is inconsistent with the effect seen for CFL1^{2KR} on F-actin. ML-792 is a selective inhibitor of the SAE enzyme, which leads to the inhibition of total SUMOylation (He X et al., 2017). Therefore, although ML-792 treatment can inhibit the SUMOylation of CFL1, theoretically increasing F-actin levels in the same way as the CFL1^{2KR} mutant, its general effect on the overall

SUMOylation status of the cell could cause a different outcome for the following reasons:

1. ML-792 acts as a total SUMOylation inhibitor, not exclusively targeting CFL1 SUMOylation. Consequently, its impact on F-actin may involve the inhibition of SUMOylation in other key proteins, influencing F-actin dynamics.
2. ML-792 may exert additional effects on cellular characteristics, such as viability and proliferation (He X et al., 2017; Biederstädt A et al., 2020), which could indirectly affect F-actin polymerization and depolymerization processes.
3. Ubc9 is the sole E2-conjugating enzyme for SUMOylation. Thus, knockdown of Ubc9 would have a similar effect as applying ML-792, inhibiting total SUMOylation. Studies have reported that the loss of Ubc9 is lethal, as the deficiency of Ubc9 and subsequent loss of SUMOylation can disrupt nuclear integrity, chromosomal segregation, and cytoplasmic division (Chang HM et al., 2020; Nacerddine K et al., 2005; He X et al., 2015). These effects may not favor the polymerization and depolymerization processes of F-actin.

Reference:

Biederstädt A, Hassan Z, Schneeweis C, Schick M, Schneider L, Muckenhuber A, Hong Y, Siegers G, Nilsson L, Wirth M, Dantes Z, Steiger K, Schunck K, Langston S, Lenhof HP, Coluccio A, Orben F, Slawska J, Scherger A, Saur D, Müller S, Rad R, Weichert W, Nilsson J, Reichert M, Schneider G, Keller U. SUMO pathway inhibition targets an aggressive pancreatic cancer subtype. *Gut*. 2020; 69(8): 1472-82.

Chang HM, Yeh ETH. SUMO: From Bench to Bedside. *Physiol Rev*. 2020; 100(4): 1599-619.

He X, Riceberg J, Pulukuri SM, Grossman S, Shinde V, Shah P, Brownell JE, Dick L, Newcomb J, Bence N. Characterization of the loss of SUMO pathway function on cancer cells and tumor proliferation. *PLoS One*. 2015; 10(4): e0123882.

He X, Riceberg J, Soucy T, Koenig E, Minissale J, Gallery M, Bernard H, Yang X, Liao H, Rabino C, Shah P, Xega K, Yan ZH, Sintchak M, Bradley J, Xu H, Duffey M, England D, Mizutani H, Hu Z, Guo J, Chau R, Dick LR, Brownell JE, Newcomb J, Langston S, Lightcap ES, Bence N, Pulukuri SM. Probing the roles of SUMOylation in cancer cell biology by using a selective SAE inhibitor. *Nat Chem Biol*. 2017;13(11):1164-71.

Nacerddine K, Lehembre F, Bhaumik M, Artus J, Cohen-Tannoudji M, Babinet C, Pandolfi PP, Dejean A. The SUMO pathway is essential for nuclear integrity and chromosome segregation in mice. *Dev Cell*. 2005; 9(6): 769-79.

Reviewer #3:

Ubiquitin is predominately linked to lysines in substrates, but can also be linked via amino termini of proteins. Whether other ubiquitin-like proteins can also be linked to amino termini of proteins is currently unclear. Here,

the authors propose that cofilin-1 is sumoylated at its amino terminus and this regulates actin depolymerization. The concept is interesting, but the findings are largely based on overexpression experiments and could therefore represent artefacts that have no physiological relevance.

We thank Reviewer #3 for this positive comment on our work and his/her constructive criticisms, insightful and thoughtful suggestions that have helped us clarify and strengthen our main findings. We have performed suggested experiments and incorporated new data in the revised manuscript.

Comments:

1. Virtually all experiments are carried out using overexpression of SUMO and overexpression of cofilin-1, which likely creates overexpression artefacts. The start of the results section is particularly weak because a potential overexpression artefact is not a sound base for the project. Mass spectrometry evidence for N-terminal sumoylation of endogenous cofilin-1 by endogenous SUMO is currently missing and is critical for the credibility of the manuscript.

Response: We appreciate the reviewer's insightful suggestions. It is indeed important to address potential overexpression artifacts and establish a solid foundation for the project. In our study, we primarily utilized overexpression of SUMO1^{E93R} in Neuro-2a cells to identify the N-terminal SUMOylation of endogenous CFL1 through mass spectrometry (MS). This method has been widely accepted and applied in previous studies, as it simplifies the MS detection of SUMO-modified sites. However, we acknowledge the concern raised by the reviewer regarding the potential artifacts associated with overexpression.

Many previous studies have successfully utilized a method involving the overexpression of SUMO plasmids carrying specific mutations, such as T95R or E93R (Knuesel M et al., 2005; Aukrust I et al., 2013; Liu H et al., 2020; Lumpkin RJ et al., 2017; Wang XD et al., 2015; Impens F et al., 2014). These mutations offer advantages in simplifying the detection of SUMO-modified sites using mass spectrometry (MS). Researchers have effectively employed these SUMO mutants in conjunction with affinity purification to identify substrates of SUMOylation.

To address this concern, we attempted an alternative approach by immunoprecipitating endogenous CFL1 in Neuro-2a cells and analyzing it for the presence of SUMO1 peptides using MS. Regrettably, we were unable to detect SUMOylated peptides of N-terminal CFL1 using this method. The relatively long peptide remnant left behind by endogenous SUMO1 (19 amino acids, ELGMEEEDVIEVYQEQTGG) combined with the N-terminal 13 amino acids of CFL1 (MASGVAVSDGVIK) should result in a peptide length of 32

amino acids. This length is unlikely to be detected by MS due to intricate fragmentation ion patterns during analysis (Swaney DL et al., 2010).

To overcome these limitations and enhance the credibility of future studies, we propose employing gene editing techniques to introduce an E93R mutation into the endogenous SUMO1 gene. This approach would avoid potential interference caused by plasmid transfection and provide a more physiologically relevant model for investigating endogenous SUMOylation. Thank you for highlighting these important points, and we appreciate your valuable feedback.

Reference:

Aukrust I, Bjørkhaug L, Negahdar M, Molnes J, Johansson BB, Müller Y, Haas W, Gygi SP, Søvik O, Flatmark T, Kulkarni RN, Njølstad PR. SUMOylation of pancreatic glucokinase regulates its cellular stability and activity. *J Biol Chem.* 2013; 288(8): 5951-62.

Impens F, Radoshevich L, Cossart P, Ribet D. Mapping of SUMO sites and analysis of SUMOylation changes induced by external stimuli. *Proc Natl Acad Sci U S A.* 2014; 111(34): 12432-7.

Knuesel M, Cheung HT, Hamady M, Barthel KK, Liu X. A method of mapping protein sumoylation sites by mass spectrometry using a modified small ubiquitin-like modifier 1 (SUMO-1) and a computational program. *Mol Cell Proteomics.* 2005; 4(10): 1626-36.

Liu H, Weng W, Guo R, Zhou J, Xue J, Zhong S, Cheng J, Zhu MX, Pan SJ, Li Y. Olig2 SUMOylation protects against genotoxic damage response by antagonizing p53 gene targeting. *Cell Death Differ.* 2020; 27(11): 3146-61.

Lumpkin RJ, Gu H, Zhu Y, Leonard M, Ahmad AS, Clauser KR, Meyer JG, Bennett EJ, Komives EA. Site-specific identification and quantitation of endogenous SUMO modifications under native conditions. *Nat Commun.* 2017; 8(1): 1171.

Swaney DL, Wenger CD, Coon JJ. Value of using multiple proteases for large-scale mass spectrometry-based proteomics. *J Proteome Res.* 2010; 9(3): 1323-9.

Wang XD, Gong Y, Chen ZL, Gong BN, Xie JJ, Zhong CQ, Wang QL, Diao LH, Xu A, Han J, Altman A, Li Y. TCR-induced sumoylation of the kinase PKC- θ controls T cell synapse organization and T cell activation. *Nat Immunol.* 2015; 16(11): 1195-203.

2. As far as I am aware no previous project has ever found that endogenous cofilin-1 is sumoylated. The authors attempt to show endogenous sumoylation of cofilin-1 in Figure 1d, but the evidence is not convincing as the extra band in the co-immunoprecipitation could be a minor degradation product of the heavy chain, or a doublet of cofilin-1 because of a cysteine-bridge. Moreover, the cofilin-1 antibody appears not fully specific. Cropping of the blots is suboptimal as the relevant parts of the cofilin-1 blots are missing in the main figures. Does the reciprocal experiment work – IP of SUMO1, blot for CFL1?

Response: We appreciate the reviewer's feedback. In response to the reviewer's suggestion, we conducted immunoprecipitation of mouse cerebral

cortex lysates using an anti-SUMO1 antibody followed by Western blotting with anti-CFL1 antibody (Reviewer Figure 10). We detected a 35-40 kD band corresponding to the expected size of SUMO1-modified CFL1. This result provides further support for the band indicated by the red arrow in the original Figure 1d as endogenous SUMOylated CFL1.

We apologize for suboptimal cropping of the blots in the main figures. Regarding the concerns raised about the band in Figure 1d, it is important to note that the band cannot be attributed to the degradation product of the heavy chain since no similar band is observed in the IgG lane. Furthermore, the band cannot be a doublet of CFL1 formed by a cysteine bridge, because the immunoprecipitation was performed under denaturing conditions, where the cysteine bridge is disrupted through the addition of dithiothreitol (DTT), β -mercaptoethanol, and heating (Emerson D et al., 1993).

Reference:

Emerson D, Ghiorse WC. Role of disulfide bonds in maintaining the structural integrity of the sheath of *Leptothrix discophora* SP-6. *J Bacteriol.* 1993; 175(24): 7819-27.

3. Modification of cofilin-1 and regulation of actin depolymerization are quite unexpected as SUMO is a nuclear protein. This raises the question of the physiological relevance of the findings. Is overexpressed SUMO in figure 4F also artificially located in the cytoplasm? The authors need to include overexpression of wild-type conjugatable and non-conjugatable SUMO with and without overexpressing cofilin-1 to test for potential overexpression artefacts of SUMO. Are endogenous cofilin-1 and actin depolymerization affected upon blocking endogenous SUMO signaling by E1 inhibiting compounds ML792 or TAK981?

Response: We thank the reviewer for the insightful comments. Regarding the SUMOylation of CFL1 and the regulation of actin depolymerization, we would like to address the physiological relevance of our findings.

1. While it is true that SUMO is primarily known as a nuclear protein, it has been demonstrated that SUMOylation can also occur on proteins in the cytoplasm. Various extra-nuclear proteins have been shown to be modified by SUMOylation, as listed in the table below:

Proteins	Locations	Reference
PTP1B	cytoplasm	Dadke S et al., Nat Cell Biol , 2007
Synapsin Ia	cytoplasm	Tang LT et al., Nat Commun , 2015
ANXA1	cytoplasm	Li X et al., Sci Adv , 2021
PDPK1	cytoplasm	Hu B et al., Autophagy , 2021
PKC α	cytoplasm	Sun H et al., Nat Commun , 2014
PKC δ	cytoplasm	Gao S et al., FEBS J , 2021
PKC ϵ	cytoplasm	Zhao X et al., Cell Rep , 2020
NEMO/IKK γ	cytoplasm	Desterro JM et al., Mol Cell , 1998

AMPK	cytoplasm	Rubio T et al., Mol Biol Cell , 2013
RIM1 α	cytomembrane	Girach F et al., Cell Rep , 2013
TRPV1	cytomembrane	Wang Y et al., Nat Commun , 2018
K ₂ P ₁	cytomembrane	Rajan S et al., Cell , 2005
K _{v1.5}	cytomembrane	Benson MD et al., Proc Natl Acad Sci U S A , 2007
K _{v7.2}	cytomembrane	Qi Y et al., Neuron , 2014

Therefore, it is not a surprise that CFL1, which is primarily located in the cytoplasm, undergoes SUMOylation.

2. In response to the reviewer's request, we performed experiments in both wild-type CHO-K1 cells and endogenous CFL1-knockdown cells expressing CFL1^{WT}-HA. We overexpressed either wild-type conjugatable SUMO1 (SUMO1^{WT}) or a non-conjugatable mutant (SUMO1^{G97A}). Consistent with the findings presented in Figure 4f, overexpression of SUMO1^{WT} promoted F-actin depolymerization, while overexpression of SUMO1^{G97A} did not (Reviewer Figure 8). This supports our conclusion that N-terminal SUMOylation of CFL1 regulates F-actin dynamics.

3. We investigated the effects of blocking endogenous SUMO signaling by using E1 inhibiting compound, ML792. In both wild-type CHO-K1 cells and endogenous CFL1-knockdown cells expressing CFL1^{WT}-HA, we observed a significant reduction in F-actin levels after treating the cells with 10 μ M of ML-792 for 24 hours (Reviewer Figure 8), which is opposite from the effect of the CFL1^{2KR} mutant on F-actin. ML-792 selectively blocks the activity of the SAE enzyme and total SUMOylation (He X et al., 2017). Therefore, although ML-792 treatment can inhibit the SUMOylation of CFL1, theoretically increasing F-actin levels in the same way as the CFL1^{2KR} mutant, its general effect on the overall SUMOylation status of the cell could cause a different outcome for the following reasons:

1. ML-792 is a broad SUMOylation inhibitor, affecting not only the SUMOylation of CFL1 but also that of other key proteins. Therefore, the effect of ML-792 on F-actin may not solely result from the de-SUMOylation of CFL1.
2. ML-792 might have additional effects on cellular characteristics such as viability and proliferation (He X et al., 2017; Biederstädt A et al., 2020), which could potentially influence the polymerization and depolymerization of F-actin.
3. As we know, Ubc9 is the sole E2-conjugating enzyme responsible for SUMOylation. Thus, knockdown of Ubc9 would be similar to the application of ML-792, inhibiting total SUMOylation. Studies have reported that the loss of Ubc9 is lethal and can lead to disruptions in nuclear integrity, chromosomal segregation, and cytoplasmic division (Chang HM et al., 2020; Nacerddine K et al., 2005; He X et al., 2015). These effects may not be conducive to the processes of F-actin

polymerization and depolymerization.

Reference:

- Benson MD, Li QJ, Kieckhafer K, Dudek D, Whorton MR, Sunahara RK, Iñiguez-Lluhí JA, Martens JR. SUMO modification regulates inactivation of the voltage-gated potassium channel Kv1.5. *Proc Natl Acad Sci U S A*. 2007; 104(6): 1805-10.
- Biederstädt A, Hassan Z, Schneeweis C, Schick M, Schneider L, Muckenhuber A, Hong Y, Siegers G, Nilsson L, Wirth M, Dantes Z, Steiger K, Schunck K, Langston S, Lenhof HP, Coluccio A, Orben F, Slawska J, Scherger A, Saur D, Müller S, Rad R, Weichert W, Nilsson J, Reichert M, Schneider G, Keller U. SUMO pathway inhibition targets an aggressive pancreatic cancer subtype. *Gut*. 2020; 69(8): 1472-82.
- Chang HM, Yeh ETH. SUMO: From Bench to Bedside. *Physiol Rev*. 2020; 100(4): 1599-619.
- Dadke S, Cotteret S, Yip SC, Jaffer ZM, Haj F, Ivanov A, Rauscher F 3rd, Shuai K, Ng T, Neel BG, Chernoff J. Regulation of protein tyrosine phosphatase 1B by sumoylation. *Nat Cell Biol*. 2007; 9(1): 80-5.
- Desterro JM, Rodriguez MS, Hay RT. SUMO-1 modification of I κ B α inhibits NF- κ B activation. *Mol Cell*. 1998; 2(2): 233-9.
- Gao S, Zhao X, Hou L, Ma R, Zhou J, Zhu MX, Pan SJ, Li Y. The interplay between SUMOylation and phosphorylation of PKC δ facilitates oxidative stress-induced apoptosis. *FEBS J*. 2021; 288(22): 6447-64.
- Girach F, Craig TJ, Rocca DL, Henley JM. RIM1 α SUMOylation is required for fast synaptic vesicle exocytosis. *Cell Rep*. 2013; 5(5): 1294-301.
- He X, Riceberg J, Pulukuri SM, Grossman S, Shinde V, Shah P, Brownell JE, Dick L, Newcomb J, Bence N. Characterization of the loss of SUMO pathway function on cancer cells and tumor proliferation. *PLoS One*. 2015; 10(4): e0123882.
- He X, Riceberg J, Soucy T, Koenig E, Minissale J, Gallery M, Bernard H, Yang X, Liao H, Rabino C, Shah P, Xega K, Yan ZH, Sintchak M, Bradley J, Xu H, Duffey M, England D, Mizutani H, Hu Z, Guo J, Chau R, Dick LR, Brownell JE, Newcomb J, Langston S, Lightcap ES, Bence N, Pulukuri SM. Probing the roles of SUMOylation in cancer cell biology by using a selective SAE inhibitor. *Nat Chem Biol*. 2017; 13(11): 1164-71.
- Hu B, Zhang Y, Deng T, Gu J, Liu J, Yang H, Xu Y, Yan Y, Yang F, Zhang H, Jin Y, Zhou J. PDPK1 regulates autophagosome biogenesis by binding to PIK3C3. *Autophagy*. 2021; 17(9): 2166-83.
- Li X, Xia Q, Mao M, Zhou H, Zheng L, Wang Y, Zeng Z, Yan L, Zhao Y, Shi J. Annexin-A1 SUMOylation regulates microglial polarization after cerebral ischemia by modulating IKK α stability via selective autophagy. *Sci Adv*. 2021; 7(4): eabc5539.
- Nacerddine K, Lehembre F, Bhaumik M, Artus J, Cohen-Tannoudji M, Babinet C, Pandolfi PP, Dejean A. The SUMO pathway is essential for nuclear integrity and chromosome segregation in mice. *Dev Cell*. 2005; 9(6): 769-79.
- Qi Y, Wang J, Bomben VC, Li DP, Chen SR, Sun H, Xi Y, Reed JG, Cheng J, Pan HL, Noebels JL, Yeh ET. Hyper-SUMOylation of the Kv7 potassium channel diminishes the M-current leading to seizures and sudden death. *Neuron*. 2014; 83(5): 1159-71.

Rajan S, Plant LD, Rabin ML, Butler MH, Goldstein SA. Sumoylation silences the plasma membrane leak K⁺ channel K2P1. *Cell*. 2005; 121(1): 37-47.

Rubio T, Vernia S, Sanz P. Sumoylation of AMPK β 2 subunit enhances AMP-activated protein kinase activity. *Mol Biol Cell*. 2013; 24(11): 1801-11, S1-4.

Sun H, Lu L, Zuo Y, Wang Y, Jiao Y, Zeng WZ, Huang C, Zhu MX, Zamponi GW, Zhou T, Xu TL, Cheng J, Li Y. Kainate receptor activation induces glycine receptor endocytosis through PKC deSUMOylation. *Nat Commun*. 2014; 5:4980.

Tang LT, Craig TJ, Henley JM. SUMOylation of synapsin Ia maintains synaptic vesicle availability and is reduced in an autism mutation. *Nat Commun*. 2015; 6:7728.

Wang Y, Gao Y, Tian Q, Deng Q, Wang Y, Zhou T, Liu Q, Mei K, Wang Y, Liu H, Ma R, Ding Y, Rong W, Cheng J, Yao J, Xu TL, Zhu MX, Li Y. TRPV1 SUMOylation regulates nociceptive signaling in models of inflammatory pain. *Nat Commun*. 2018; 9(1): 1529.

Zhao X, Xia B, Cheng J, Zhu MX, Li Y. PKC ϵ SUMOylation Is Required for Mediating the Nociceptive Signaling of Inflammatory Pain. *Cell Rep*. 2020; 33(1): 108191.

4. Most amino termini of proteins are normally heavily acetylated. Is the N-terminus of endogenous cofilin-1 also acetylated? How is the acetyl group replaced by SUMO? The ‘surprise’ result that overexpressed cofilin-1 does not start at methionine, but at alanine is not trustworthy, since it could also represent an overexpression artefact.

Response: We appreciate the reviewer's valuable comment. In Supplementary Figure 5a, we show that endogenous CFL1 can undergo N-terminal acetylation when co-overexpressed with N-terminal acetyltransferase following N-terminal methionine excision (NME).

Based on our current understanding, endogenous CFL1 can undergo two distinct post-translational modifications. Firstly, N-terminal SUMOylation takes place at the N-terminal methionine residue (Met1) independently of NME. Secondly, N-terminal acetylation occurs at the second alanine residue (Ala2) and is dependent on NME. Therefore, there is currently no evidence to suggest that N-acetylation of CFL1 can be replaced by N-SUMOylation.

It has been previously reported that when various proteins are overexpressed from plasmids, they can undergo NME and subsequently be N-terminally acetylated (Hwang CS et al., 2010; Shemorry A et al., 2013). Hence, it is possible that overexpressed CFL1, following NME, may initiate at alanine.

However, we acknowledge the concern that the observation of CFL1 starting at alanine in our study could potentially be an artifact resulting from overexpression. Further investigations are needed to confirm the exact N-terminal modification patterns of endogenous CFL1 and to determine the functional implications of these modifications.

Reference:

Hwang CS, Shemorry A, Varshavsky A. N-terminal acetylation of cellular proteins creates specific degradation signals. *Science*. 2010; 327(5968): 973-7.

Shemorry A, Hwang CS, Varshavsky A. Control of protein quality and stoichiometries by N-terminal acetylation and the N-end rule pathway. *Mol Cell*. 2013; 50(4): 540-51.

5. Currently, it is unclear whether cofilin-1 is the only protein potentially regulated by N-terminal sumoylation. It would be interesting to use an unbiased approach to investigate whether other proteins are regulated in a similar manner – using endogenous SUMO and endogenous SUMO target proteins.

Response: We thank the reviewer for bringing up this interesting point. In order to address this question, we conducted immunoprecipitation of endogenous SUMO1 in Neuro-2a cells and performed mass spectrometry (MS) analysis to detect N-terminal peptides of endogenous SUMO1. However, we were unable to detect any SUMOylated N-terminal peptides. We believe that the following reasons may account for this result:

1. The low abundance and dynamic nature of SUMOylation pose significant challenges for the identification of SUMOylated substrates through mass spectrometry (Becker J et al., 2013; Vertegaal ACO, 2022). We speculate that N-terminal SUMOylation might represent a unique form of SUMOylation, and its abundance could be extremely low compared to other SUMOylation sites.
2. The theoretical length of the SUMO1-ylated N-terminal peptide is quite long, considering the 19 amino acids (ELGMEEEDVIEVYQEQTGG) from the endogenous SUMO1 added to the N-terminal peptide of the substrate, which has to have a certain length in order to be identifiable based on mass. The long length makes it very difficult for mass spectrometry techniques to make identification of the peptide (Swaney DL et al., 2010).

Further investigations using alternative approaches are needed to determine whether any other proteins are also N-terminal SUMOylation as CFL1.

Reference:

Becker J, Barysch SV, Karaca S, Dittner C, Hsiao HH, Berriel Diaz M, Herzig S, Urlaub H, Melchior F. Detecting endogenous SUMO targets in mammalian cells and tissues. *Nat Struct Mol Biol*. 2013; 20(4): 525-31.

Swaney DL, Wenger CD, Coon JJ. Value of using multiple proteases for large-scale mass spectrometry-based proteomics. *J Proteome Res*. 2010; 9(3): 1323-9.

Vertegaal ACO. Signalling mechanisms and cellular functions of SUMO. *Nat Rev Mol Cell Biol*. 2022; 23(11):715-31.

Additional figures for the reviewers

Reviewer Figure 1. Different centrifugation conditions do not significantly affect the outcomes of the assay. The F-actin was ultracentrifuged with four different ultracentrifugation conditions including 100,000 g for 30 min, 100,000 g for 60 min, 220,000 g for 30 min, and 440,000 g for 30 min. The supernatants (P) and pellets (S) were separately subjected to SDS-PAGE and stained with CBB. Black arrowhead indicates actin.

Reviewer Figure 2. Flow chambers (red frame) assembled based on descriptions in the literature.

Reviewer Figure 3. The comparison of cofilin-induced F-actin depolymerization between Figure 4e and literatures. The F-actin depolymerization curve (green) measured by Nadkarni and Brieher (2014) and pyrenyl actin quenching curve (gray) measured by Nadkarni and Brieher (2014) were overlaid on Figure 4e.

Reviewer Figure 4. Endogenous CFL1 can be SUMOylated in CHO-K1 cells. Lysates prepared from CHO-K1 cells under denaturing conditions were subjected to IP with anti-CFL1 antibody, followed by IB with anti-SUMO1 and anti-CFL1 antibodies. The original lysates were analyzed by IB with anti-CFL1 antibodies for input, and anti-GAPDH antibody for loading control. Arrow indicates endogenous SUMOylated CFL1.

Reviewer Figure 5. The N-terminus of CFL1 is modified by SUMO2. (a) Lysates from CHO-K1 cells transiently transfected with empty vector (-), CFL1-HA, His-SUMO2, RGS-SENP2, and RGS-SENP2^{CS} at various combinations as indicated for 24 hours were subjected to De-IP with the anti-HA antibody, which was followed by IB using anti-SUMO2/3 and anti-HA antibodies. The original lysates were analyzed by IB with anti-HA and anti-SUMO2/3 antibodies for input, and anti-GAPDH antibody for loading control. (b) Lysates from CHO-K1 cells transiently transfected with empty vector (-), CFL1^{25KR}-HA, His-SUMO2, and Myc-Ubc9 at various combinations as indicated for 24 hours were subjected to De-IP and IB as in Reviewer Fig. 4a. The lysine-less 25KR mutant was still SUMO2 conjugated, which was further enhanced by co-expressing Ubc9. (c) Lysates from CHO-K1 cells transiently transfected with empty vector (-), CFL1^{WT}-HA, His-SUMO2, and Myc-Naa60 at various combinations as indicated for 24 hours were subjected to De-IP and IB as in Reviewer Fig. 4a. Coexpression of Naa60 abolished CFL1 SUMOylation. (d) Purified wide-type CFL1 and GST-CFL1 proteins were subject to *in vitro* SUMOylation as in Fig. 2f, and then incubated with thrombin at room temperature for 12 hours. The reaction was terminated by diluting in SDS loading buffer. The samples were analyzed by IB with anti-SUMO2/3, anti-CFL1 and anti-GST antibodies.

Reviewer Figure 6. SUMOylated CFL1 can be detected by anti-His antibody. Lysates from CHO-K1 cells transiently transfected with empty vector (-), CFL1-HA, His-SUMO1, RGS-SENP1, and RGS-SENP1^{CS} at various combinations as indicated for 24 hours were subjected to De-IP with the anti-HA antibody, which was followed by IB using anti-His and anti-HA antibodies. The original lysates were analyzed by IB with anti-HA and anti-SUMO1 antibodies for input, and anti-GAPDH antibody for loading control.

Reviewer Figure 7. CFL1 interacted with E1 enzyme subunits and SENP1. Proximity ligation assay (PLA) of CFL1 with SAE1, SAE2 and SENP1 in CHO-K1 cells suggests that CFL1 comes into close proximity with the E1 enzyme subunits and SENP1, respectively (red dots). Nuclei were stained with DAPI (blue). Scale bar = 50 μ m.

Reviewer Figure 8. The SUMOylation level of CFL1^{25KR} was lower than CFL1^{WT}. Lysates from CHO-K1 cells transiently transfected with empty vector (-), CFL1^{WT}-HA, CFL1^{25KR}-HA and His-SUMO1 at various combinations as indicated for 24 hours were subjected to De-IP with the anti-HA antibody, which was followed by IB using anti-SUMO1 and anti-HA antibodies. The original lysates were analyzed by IB with anti-HA and anti-SUMO1 antibodies for input, and anti-GAPDH antibody for loading control.

Reviewer Figure 9. The effect of SUMO1^{WT}, SUMO1^{G97A} and ML-792 on F-actin. Wild type CHO-K1 cells (left 4 panels), and CHO-K1 cells with the endogenous CFL1 stably knocked down and WT CHO-K1 reintroduced by transfection (right 3 panels) were used. Cells were co-transfected with the indicated SUMO1^{WT} or SUMO1^{G97A}, or treated with ML-792 (10 μ M, 24 hours). Cells were identified by immunostaining with mouse anti-CFL1 antibody followed by goat anti-mouse antibody conjugated with Alexa 488 (green). Cells

were stained for F-actin using rhodamine-conjugated phalloidin (*red*). Scale bar = 50 μ m.

Reviewer Figure 10. SUMO1 conjugation of endogenous CFL1 in mouse cerebral cortex *in vivo*. Lysates prepared from mouse cerebral cortices under denaturing conditions were subjected to IP with anti-SUMO1 antibody, followed by IB with anti-SUMO1 and anti-CFL1 antibodies. The original lysates were analyzed by IB with anti-CFL1 antibodies for input, and anti-GAPDH antibody for loading control. Arrow indicates endogenous SUMOylated CFL1.

We thank all reviewers and editors for the constructive comments on our work. We have addressed all the comments and suggestions. We believe that the new data added in our revised manuscript provide additional strong support for our conclusions and substantially strengthen the paper. We hope that our manuscript is now acceptable for publication in *Nature Communications*.

Thank you very much for your consideration.

Yours sincerely,

Yong Li, Ph.D.
Investigator and Head,
Laboratory of Receptor Trafficking and Diseases
Department of Biochemistry and Molecular Cell Biology
Shanghai Jiao Tong University School of Medicine
280 South Chongqing Road
Shanghai 200025, China
Tel: 86-21-64665820
Fax: 86-21-64661525
E-mail: liyong68@shsmu.edu.cn
<http://www.shsmu.edu.cn>

REVIEWERS' COMMENTS

Reviewer #1 (Remarks to the Author):

Summary of Critique: In this revised version of their original submission from 2022, the authors address every point of the three reviewers, some more substantially than others. It is quite clear that Cofilin-1, the ubiquitously expressed mammalian isoform of the ADF/cofilin family of actin regulatory proteins, is here shown to be a substrate for SUMOylation, although the amounts of the SUMO form are so low that it is only observed by overexpression of SUMO. However, the finding that a non-conjugatable mutant of SUMO did not alter actin assembly levels found with conjugatable SUMO does support the importance of the cofilin-modification to its activity. The N-terminal site of SUMOylation of cofilin on the alpha-amino group of N-terminal met is very well characterized. Because cofilin is rapidly demethylated and N-acetylated, the results of this study strongly suggest that SUMOylation is associated with newly synthesized cofilin, which should certainly be addressed in any future study. SUMOylated cofilin binds somewhat better to F-actin than WT cofilin using in vitro co-sedimentation assays.

Critique

Several of the specific points raised in the first review have been addressed by additional experiments utilizing alternative methods with results that support the original observations. The inclusion of quantitative data from triplicate immunoblots also helps to strengthen data interpretation. The authors did attempt to address every issue raised by reviewers but were not successful in all instances.

Some reviewers questioned how the primarily nuclear localized SUMO would function with a cytoplasmic cofilin, and although the authors pointed out several studies on SUMOylation of cytoplasmic proteins, they failed to point out that cofilin has a bipartite nuclear localization sequence and is thought to be a major transporter of actin into the cell nucleus. In addition, the authors might want to cite a recent publication in which cables of cofilin-actin (phalloidin negative) were discovered to wind around the internal surface of the nuclear envelope (doi: 10.1111/gtc.12930. Epub 2022 Mar 6.).

The authors state in their rebuttal letter that: "Hylton et al. analyzed the effects of different fixing reagents and different permeabilizing reagents on phalloidin staining and cofilin staining. They found that PFA fixation + Triton X-100 permeabilization disrupts cofilin-actin bundle staining but does not affect phalloidin staining of F-actin; PFA fixation + acetone permeabilization slightly inhibits the phalloidin signal; while methanol fixation disrupts phalloidin binding to F-actin (Hylton RK et al., 2022). Therefore, there appears to be no absolute certainty that any of the fixation methods is suitable for detecting cofilin-actin." However, this conclusion is unwarranted since cofilin-actin filaments are preserved after PFA fixation and permeabilization with -20°C methanol. Bundles of cofilin-actin, labeled through expression of an mRFP-cofilin, could be fixed with PFA and immunolabeled with a cofilin-antibody after the methanol treatment but not when Triton was used (DOI: 10.1007/978-1-0716-2811-9_18). Bundles never labeled with phalloidin following either methanol or Triton permeabilization which shows that phalloidin staining is not a good measure of assembled actin unless one is excluding cofilin-saturated actin filaments from the F-actin pool. However, these bundles will contribute to the sedimented pool of F-actin as quantified by centrifugation. Thus, the phalloidin binding as a measure of assembled actin is much less reliable than the sedimentation assays performed by the authors. Kinetic studies show SUMOylation of cofilin enhances its depolymerizing activity.

It is unfortunate that the in vitro studies on actin filament severing/depolymerization did not work but as the authors point out, these are designed for a TIRF microscope to which the authors had no access. It is also disappointing that more cell biological studies directed at showing SUMOylation of cofilin had some dramatic effect on actin dynamics in cells could not be included. However, the data that is provided is certainly important and well done and can serve as the basis for future work where local cofilin synthesis (such as in dendritic spines) could be evaluated for SUMO effects on local spine architecture.

Reviewer #2 (Remarks to the Author):

The authors have now adequately addressed my original concerns in the revised manuscript. I have only a few minor comments:

- 1) Quantifications of the western blot or IF results from many different figures are shown in Supplementary Figure 6, as a separate figure. Doing so makes it unnecessarily difficult to follow the figures and disrupts the flow. In my opinion, it would be better if each WB has its corresponding quantification next to it in the relevant figure. The figures can certainly be rearranged to do this.
- 2) Reviewer Figure 6 is convincing and should be included in the manuscript.
- 3) Reviewer Figure 7 also contains important piece of data. If these results are reproducible, the authors should quantify the PLA signals and include it as a figure in the manuscript.

Reviewer #3 (Remarks to the Author):

The authors propose that cofilin-1 is sumoylated at its amino terminus and this regulates actin depolymerization. The concept is interesting, but the findings are largely based on overexpression experiments and could therefore represent artefacts that have no physiological relevance. Unfortunately, the revision experiments to address this key point have largely failed. My concerns that the results could represent overexpression artefacts that have no physiological relevance remain.

Authors' Responses to the Reviewers' Comments:

Reviewer #1 (reviewers' comments in *italic*):

Summary of Critique: In this revised version of their original submission from 2022, the authors address every point of the three reviewers, some more substantially than others. It is quite clear that Cofilin-1, the ubiquitously expressed mammalian isoform of the ADF/cofilin family of actin regulatory proteins, is here shown to be a substrate for SUMOylation, although the amounts of the SUMO form are so low that it is only observed by overexpression of SUMO. However, the finding that a non-conjugatable mutant of SUMO did not alter actin assembly levels found with conjugatable SUMO does support the importance of the cofilin-modification to its activity. The N-terminal site of SUMOylation of cofilin on the alpha-amino group of N-terminal met is very well characterized. Because cofilin is rapidly demethionated and N-acetylated, the results of this study strongly suggest that SUMOylation is associated with newly synthesized cofilin, which should certainly be addressed in any future study. SUMOylated cofilin binds somewhat better to F-actin than WT cofilin using in vitro co-sedimentation assays.

Response: We appreciate Reviewer #1 for her/his positive feedback and constructive comments that have greatly improved our manuscript.

Critique: *Several of the specific points raised in the first review have been addressed by additional experiments utilizing alternative methods with results that support the original observations. The inclusion of quantitative data from triplicate immunoblots also helps to strengthen data interpretation. The authors did attempt to address every issue raised by reviewers but were not successful in all instances.*

Response: We are grateful for the reviewer's insightful evaluation of our manuscript. Your feedback is valuable to us. We have indeed addressed specific concerns by conducting alternative experiments and presenting quantified data, which has further fortified our findings. While we endeavored to tackle all raised issues, we encountered certain limitations. Your input has significantly contributed to the enhancement of our manuscript.

Some reviewers questioned how the primarily nuclear localized SUMO would function with a cytoplasmic cofilin, and although the authors pointed out several studies on SUMOylation of cytoplasmic proteins, they failed to point out that cofilin has a bipartite nuclear localization sequence and is thought to be a major transporter of actin into the cell nucleus. In addition, the authors might want to cite a recent publication in which cables of cofilin-actin (phalloidin negative) were discovered to wind around the internal surface of the nuclear envelope (doi: 10.1111/gtc.12930. Epub 2022 Mar 6.).

Response: We are grateful for the thorough and insightful feedback provided by the reviewer. Your observations are greatly appreciated. We acknowledge the significance of the bipartite nuclear localization sequence in cofilin and its role as a potential transporter of actin into the nucleus. Furthermore, we appreciate your suggestion to reference the recent publication that unveils the presence of cofilin-actin cables winding around the internal surface of the nuclear envelope (doi: 10.1111/gtc.12930. Epub 2022 Mar 6.). These additions have been thoughtfully integrated into the revised manuscript, and your input has significantly contributed to the enhancement of our study.

The authors state in their rebuttal letter that: “Hylton et al. analyzed the effects of different fixing reagents and different permeabilizing reagents on phalloidin staining and cofilin staining. They found that PFA fixation + Triton X-100 permeabilization disrupts cofilin-actin bundle staining but does not affect phalloidin staining of F-actin; PFA fixation + acetone permeabilization slightly inhibits the phalloidin signal; while methanol fixation disrupts phalloidin binding to F-actin (Hylton RK et al., 2022). Therefore, there appears to be no absolute certainty that any of the fixation methods is suitable for detecting cofilin-actin.” However, this conclusion is unwarranted since cofilin-actin filaments are preserved

after PFA fixation and permeabilization with -20o C methanol. Bundles of cofilin-actin, labeled through expression of an mRFP-cofilin, could be fixed with PFA and immunolabeled with a cofilin-antibody after the methanol treatment but not when Triton was used (DOI: 10.1007/978-1-0716-2811-9_18). Bundles never labeled with phalloidin following either methanol or Triton permeabilization which shows that phalloidin staining is not a good measure of assembled actin unless one is excluding cofilin-saturated actin filaments from the F-actin pool. However, these bundles will contribute to the sedimented pool of F-actin as quantified by centrifugation. Thus, the phalloidin binding as a measure of assembled actin is much less reliable than the sedimentation assays performed by the authors. Kinetic studies show SUMOylation of cofilin enhances its depolymerizing activity.

Response: We have carefully considered the feedback provided by the reviewer and incorporated the appropriate modifications into our manuscript. We acknowledge the concerns raised by the reviewer and deeply value their valuable insights. In response to their observations, we have introduced a new passage in the Discussion section to highlight the limitations associated with phalloidin staining when utilizing PFA fixation + Triton X-100 permeabilization: “Third, the use of paraformaldehyde fixation and Triton X-100 permeabilization in the in vivo F-actin staining assay can disrupt cofilin-actin bundle staining⁴⁰. As a result, relying solely on phalloidin staining may not offer an optimal approach for assessing polymerized actin. To delve deeper into the dynamic regulation of F-actin by CFL1 SUMOylation, we intend to explore alternative and more effective methodologies.”

Reviewer #2:

The authors have now adequately addressed my original concerns in the revised manuscript.

Response: We thank Reviewer #2 for the support and recommendation to accept our paper. We appreciate the reviewer's acknowledgment of our efforts in addressing their initial concerns within the revised manuscript. Your valuable feedback and suggestions have been instrumental in enhancing the clarity and strength of our findings.

I have only a few minor comments:

1) Quantifications of the western blot or IF results from many different figures are shown in Supplementary Figure 6, as a separate figure. Doing so makes it unnecessarily difficult to follow the figures and disrupts the flow. In my opinion, it would be better if each WB has its corresponding quantification next to it in the relevant figure. The figures can certainly be rearranged to do this.

Response: We appreciate the reviewer's insightful suggestions. In the revised manuscript, we have reorganized the figures so that the quantitative statistical graphs are positioned alongside their respective Western blot or immunofluorescence data.

2) Reviewer Figure 6 is convincing and should be included in the manuscript.

Response: We thank the reviewer for their input. The figure has been incorporated into the revised manuscript as Supplementary Figure 1c.

3) Reviewer Figure 7 also contains important piece of data. If these results are reproducible, the authors should quantify the PLA signals and include it as a figure in the manuscript.

Response: We highly value the reviewer's input. In response, we have incorporated the findings into the revised manuscript as Supplementary Figure 6.

Reviewer #3:

The authors propose that cofilin-1 is sumoylated at its amino terminus and this regulates actin depolymerization. The concept is interesting, but the findings are largely based on overexpression experiments and could therefore represent artefacts that have no physiological relevance. Unfortunately, the revision experiments to address this key point have largely failed. My concerns that the results could represent overexpression artefacts that have no physiological relevance remain.

Response: We are grateful for the reviewer's feedback and their valid apprehension regarding potential overexpression artifacts within our study. In response, we have taken measures to temper the scope of our claims within the revised manuscript, duly recognizing the limitations inherent to our revised experiments in providing a comprehensive resolution to this concern.

To bolster the physiological significance of our findings, we propose the implementation of a gene editing approach, specifically through K34 knock-in mice, as a viable alternative to the intricate N-terminal knock-in strategy. This proposed avenue aims to establish a biologically authentic model capable of effectively validating our findings while mitigating potential artifacts associated with plasmid transfection.

Therefore, we have included the following in the Discussion section to address the limitations of our study: “Finally, it's worth noting that our partial findings rely on overexpression experiments, which may not completely eliminate the potential for artifacts. Moving forward, it becomes imperative to establish a biologically authentic model that can provide more robust insights into the

precise functions of N-terminal SUMOylation on CFL1.”

We extend our gratitude to the reviewer for their insightful input, which has effectively steered us towards a more robust and credible trajectory for future investigations.

We hope that the revised manuscript with minor text edits is satisfactory and that you will formally accept this version for publication in *Nature Communications*.

Thank you very much for your consideration.

Yours sincerely,

Yong Li, Ph.D.
Investigator and Head,
Laboratory of Receptor Trafficking and Diseases
Department of Biochemistry and Molecular Cell Biology
Institute of Medical Sciences
Shanghai Jiao Tong University School of Medicine
280 South Chongqing Road
Shanghai 200025, China
Tel: 86-21-64665820
Fax: 86-21-64661525
E-mail: liyong68@shsmu.edu.cn
<http://www.shsmu.edu.cn>